



# Generating porosity during olivine carbonation via dissolution channels and expansion cracks

Tiange Xing[1], Wenlu Zhu[1], Florian Fusseis[2], Harrison Lisabeth [1,3]

[1] Department of Geology, University of Maryland, College Park, 20742, USA

[2] School of Geosciences, University of Edinburgh, Edinburgh, EH9 3FE, UK

[3] Department of Geophysics, Stanford University, Stanford, 94305, USA

*Correspondence to*: Tiange Xing (tiange@umd.edu)

**Abstract.** The olivine carbonation reaction, in which carbon dioxide is chemically incorporated to form carbonate, is central to the emerging carbon sequestration method using ultramafic rocks. The rate of this retrograde metamorphic reaction is
controlled, in part, by the available reactive surface area: as the solid volume increases during carbonation, the feasibility of this method ultimately depends on the maintenance of porosity and the creation of new reactive surfaces. We conducted in-situ dynamic x-ray microtomography and nanotomography experiments to image and quantify the porosity generation during olivine carbonation. We designed a sample setup that included a thick-walled cup (made of porous olivine aggregates with a mean grain size of either ~5 or ~80 μm) filled with loose olivine sands with grain sizes of 100-500 μm. The whole sample
assembly was reacted with a $NaHCO_3$ aqueous solution at 200 °C, under a constant confining pressure of 13 MPa and a pore pressure of 10 MPa. Using synchrotron-based X-ray microtomography, the 3-dimensional (3-D) pore structure evolution of the carbonating olivine cup was documented until the olivine aggregates became disintegrated. The dynamic microtomography data show a volume reduction in olivine at the beginning of the reaction, indicating a vigorous dissolution process consistent with the disequilibrium reaction kinetics. In the olivine cup with a grain size of ~80 μm (coarse-grained cup), dissolution
fractures developed within 30 hours, before any precipitation was observed. In the experiment with the olivine cup of ~5 μm mean grain size (fine-grained cup), idiomorphic magnesite crystals were observed on the surface of the olivine sands. The magnesite shows a near constant growth throughout the experiment, suggesting that the reaction is self-sustained. Large fractures were generated as reaction proceeds and eventually disintegrate the aggregate after 140 hours. Detailed analysis show that these are expansion cracks caused by the volume mismatch between the expanding interior and the nearly constant surface.
Nanotomography images of the reacted olivine cup reveal pervasive etch-pits and worm-holes in the olivine grains. We interpret this perforation of the solids to provide continuous fluid access, which is likely key to the complete carbonation observed in nature. Reactions proceeding through the formation of nano- to micron-scale dissolution channels provide a viable microscale mechanism in carbon sequestration practices. For the natural peridotite carbonation, a coupled-mechanism of dissolution and reaction-induced fracturing should account for the observed self-sustainability of the reaction.

**1 Introduction**

Mantle peridotites are exposed widely on the Earth's surface in tectonic settings such as mid-ocean ridges, subduction zones and ophiolites (Escartín et al., 1997; Fryer et al., 1995). Peridotite is mainly composed of olivine which is unstable at temperature below 700 °C in the presence of water (Evans, 1977), and below 500 °C in the presence of CO2-rich fluids (Johannes, 1969). The transformation of olivine to serpentine and carbonates due to fluid-rock interaction is extensively
observed in peridotite outcrops (e.g. Beinlich et al., 2012; Falk and Kelemen, 2015; Hansen et al., 2005). Rock deformation





experiments have demonstrated that fluid alteration to peridotite can strongly affect the strength and tectonics of the oceanic lithosphere (e.g. Escartín et al., 2001; Moore et al., 1996). Therefore, study of the olivine-fluid interaction is of great importance in understanding the alteration process of peridotite in a variety of tectonic regions. General peridotite alteration reaction can be formulated as follow (Hansen et al., 2005; Kelemen and Matter, 2008):

Olivine + $CO_2$ in fluid = Magnesite + Quartz,
$$Mg_2SiO_4 + 2CO_2 = 2MgCO_3 + SiO_2, \tag{1}$$

Olivine + $CO_2$ in fluid + $H_2O$ = Talc + Magnesite,
$$4Mg_2SiO_4 + 5CO_2 + H_2O = Mg_3Si_4O_{10}(OH)_2 + 5MgCO_3, \tag{2}$$

Olivine + $H_2O$ = Serpentine + Brucite,
$$2Mg_2SiO_4 + 3H_2O = Mg_3Si_2O_5(OH)_4 + Mg(OH)_2, \tag{3}$$

Brucite + $CO_2$ = Magnesite + $H_2O$,
$$Mg(OH)_2 + CO_2 = MgCO_3 + H_2O, \tag{4}$$

Although peridotite weathering reactions occur widely in nature, the rate of olivine mineralization at subsurface conditions is debated. Since the retrograde metamorphic reactions are kinetically fast, the extent of transformation is limited by fluid supply
which depends on the accessible fluid pathways. As the hydration and carbonation of olivine results in an up to ~44% increase in solid molar volume (Goff and Lackner, 1998; Hansen et al., 2005; Kelemen and Matter, 2008), carbonation of olivine is generally assumed to be self-limiting: the reaction products would gradually fill up the pore space and lead to a decrease in the porosity (Emmanuel and Berkowitz, 2006; Hövelmann et al., 2012), which in turn lowers permeability and reduces fluid supply. This negative feedback would ultimately force the alteration to cease. However, naturally occurring completely
carbonated peridotites are evidence that these limitations can be overcome. For instance, listvenite is the natural completely carbonated product of peridotite, which is composed of magnesite, quartz and trace minerals (Beinlich et al., 2012; Nasir et al., 2007). This creates a conundrum of how the large extent carbonation can be achieved with the potential self-limitation of reducing fluid pathways.

In order to explain this discrepancy between the theory and the observation, numerous studies have been conducted aiming to
find a mechanism to maintain the access for reacting fluid during olivine alteration reactions. In 1985, Macdonald and Fyfe examined naturally altered peridotite and proposed that the large volume change associated with the reaction could generate high local stresses and strains, which would cause episodic cracking. This idea has then been applied to olivine carbonation by Kelemen and Matter (2008), who proposed a positive feedback loop where fractures could be generated during the volume-expanding reaction, porosity and permeability can be maintained or even increased, which in turn would accelerate the
carbonation processes (cf. Rudge et al., 2010). In 2011, Kelemen et al. showed that in natural peridotites cross-cutting hierarchical fracture networks filled by syn-kinematic carbonate and quartz veins extend to microscopic scales. These cross-cutting networks indicate coeval carbonate crystallization and fracturing. Several studies also showed that the forces generated by the volume increase should be enough to fracture peridotite (Iyer et al., 2008; Jamtveit et al., 2009, 2011; Ulven et al., 2014).

While reaction-induced fracturing is accepted as a way to maintain fluid access, the mechanical details of the process are poorly understood. As for the mechanism that generates stresses, 'crystallization pressure' (also termed 'force of crystallization') has been proposed (e.g. Scherer, 2004; Weyl, 1959; Winkler and Singer, 1972). In this model, the



precipitation/crystallization of reaction products exert pressure around the growing crystals, and fracturing takes place when that pressure exceeds the local minimum principal stress (Kelemen and Hirth, 2012). Salt crystallization (Scherer, 2004) is a
common example where high crystallization forces due to the nucleation of precipitates in pore space cause samples to 'burst from the inside' (see Figure 1a). However, studies have shown that the crystallization force is low in the olivine carbonation system (e.g. van Noort et al., 2017). Because of the lack of experimental evidence of crystallization forces during olivine mineralization, Zhu et al. (2016) proposed an 'expansion cracking' mechanism as an alternative model after successfully producing reaction-induced fractures in an in-situ synchrotron x-ray microtomography study. In the 'expansion cracking'
model, tensile stresses are generated due to the volume mismatch between regions with different precipitation rates, leading to cracks forming in regions that expand slower than their surroundings (see Figure 1b).

Beyond fracturing, dissolution has been recognized as an important part of the olivine alteration process (e.g. Velbel, 2009; Velbel and Ranck, 2008; Wilson and Jones, 1983) and proposed as a mechanism to explain the observed complete carbonation of peridotite. In 1978, Grandstaff showed that dissolution could significantly increase the surface area through etch-pitting.
Wilson (2004) suggested that the weathering of olivine is controlled by etch-pitting and channel formation due to preferential dissolution, which assists the migration of fluid and promote further reaction. Andreani et al. (2009) suggested that permeability may be maintained during peridotite carbonation by the development of preferential flow zones. Lisabeth et al. (2017a, 2017b) observed relevant structures in dunite samples that have been reacted under controlled stress conditions, and interpreted them as a pattern of secondary porosity bands formed by dissolution coupled to locally intensified compressional
stresses.

Previous investigations of olivine carbonation were largely based on the interpretation of naturally deformed samples (e.g. Macdonald and Fyfe, 1985), thermodynamic modelling (e.g. Kelemen and Hirth, 2012) or comparison with reaction system other than olivine (e.g. leucite to analcime in Jamtveit et al., 2009). While these approaches led to significant advancements, there are limitations to the understanding of the mechanisms responsible for porosity generation during olivine carbonation
that these approaches can provide. The history of natural fault rocks is inevitably complex, and thermodynamic arguments and numerical models can only indicate a potential, while the actual progress of chemical reactions is strongly affected by interfacial structures, which vary considerably in different mineral systems. Thus, it is critical to complement such studies with laboratory experiments on olivine carbonation.

Synchrotron-based x-ray tomography is an advanced non-destructive method to capture three-dimensional images of materials.
Where processes affecting these materials are followed through time, a 4-dimensional (3 spatial dimensions + time) dataset is captured. By using x-ray transparent reaction cells (Fusseis et al., 2014b), the technique enables the investigation of fluid-rock interaction at controlled and geologically relevant conditions. We examined the carbonation process of olivine on the basis of 4-dimensional images acquired by x-ray microtomographic imaging with synchrotron radiation at the Advanced Photon Source. In this current study, we conducted a new experiment using an olivine aggregate with larger grain size (80-100 μm
compared to 0-20 μm in Zhu et al., 2016). We also performed advanced 3D analyses and quantification of the micro- and nano-tomography data obtained by Zhu et al. (2016). In an advancement of the results presented by Zhu et al. (2016), here we present direct evidence for the coupled mechanisms of dissolution and precipitation-driven fracturing during olivine carbonation and demonstrate their importance in sustaining the reaction progress at different spatial and temporal scales. We further show direct evidence of how reaction-induced fracturing operates, i.e. how stress is generated through volume-increasing reactions. A
better understanding of olivine carbonation directly applies to the geological sequestration of $CO_2$ (Gislason et al., 2010; Mani et al., 2008). The principle of in-situ carbon mineralization is the conversion of silicate and hydroxide minerals to form carbonate minerals as a stable sink for $CO_2$ (Power et al., 2013). Peridotite, because of its wide occurrence and high reactivity, is considered one of the best potential feedstocks for $CO_2$ mineralization (Andreani et al., 2009; Beinlich and Austrheim,



2012); the estimated rate of $CO_2$ consumption peridotite carbonation could be as high as $2\times10^9$ tons·km$^{-3}$ per year (Kelemen
and Matter, 2008). As the dominant constituent of peridotite, olivine becomes the most important mineral for $CO_2$ mineralization. Our study provides new insights into carbon sequestration using ultramafic rocks, and our findings on the mechanism of fracture generation during olivine carbonation could provide guidance to industrial applications.

## 2 Experimental Setup

### 2.1 Sample Configuration

The sample assembly consists of a millimeter-sized cup made from forsterite, filled with loose olivine sand (grain size 100-500 μm, see Figure 2a). In Zhu et al. (2016), a fine-grained (grains sizes between 0-20 μm) cup was used for the sample. This assembly is referred to as fine-grained olivine aggregate in the following discussion. We conducted a new in-situ microtomography experiment, in which a coarse-grained olivine cup (80-100 μm) was used. This experiment is referred to as coarse-grained olivine aggregate in the discussion. In Zhu et al. (2016), the contrast in grain size between the loose grains and
the cup wall aggregate is hypothesized as the cause of the non-uniform precipitation which is crucial to the generation of fractures in the experiment. Here, we use coarse-grained olivine aggregate in the cup wall to reduce the contrast in grain size between the cup wall and the fillings and further test their hypothesis.

The cup, which was fabricated by hot-pressing in a procedure described in Zhu et al. (2016), has inner and outer diameters of 1 and 1.8 mm respectively, with a resulting wall thickness of 0.4 mm. It has an initial porosity of 10%. The loose grains inside
the cup allowed the inspection of magnesite growth on free olivine surfaces. The sample assemblies (i.e., olivine cup + loose grains) were jacketed and loaded into an x-ray transparent pressure cell (Figure 3). A confining pressure of 13 MPa and a pore fluid of $NaHCO_3$ aqueous solution (1.5 mol·L$^{-1}$) at 10 MPa were applied to the sample. The pressure cell was then heated to 200 °C to initiate the reaction. These conditions were kept constant during the entire microtomography experiment.

### 2.2 Micro- and Nano-tomography

Third-generation synchrotron facilities produce electromagnetic radiation bright enough to allow rapid imaging even inside experimental vessels, thereby enabling studies of dynamic processes ranging over periods from seconds to days, while acquiring individual 3-dimensional (3-D) data sets in fractions of a second. Synchrotron x-ray microtomography has therefore become one of the most powerful tools in structural geology and rock mechanics studies (see Fusseis et al., 2014a for a review and Bedford et al., 2017 for a recent application).

In this experiment, synchrotron-based x-ray absorption microtomography has been used to record the dynamic carbonation of olivine in 4 dimensions. We used an x-ray transparent cell (Fusseis et al., 2014b), mounted in the upstream experimental station at beamline 2BM of the Advanced Photon Source of Argonne National Laboratory, 25 m from the source. There, a polychromatic beam filtered by 1 mm aluminum, 15 mm silicon and 8 mm borosilicate glass yielded a photon flux with an energy peak at 65 KeV (Zhu et al., 2016). A Cooke pco.edge sCMOS camera with 2560×2160 pixels (pixel size 6.5×6.5 μm$^2$)
was used in a flying scan mode. The sample-detector distance was 300 mm, which introduced a clear phase contrast signal to the data (Cloetens et al., 1996). The camera recorded projections from a 10 μm thick LuAG:Ce single crystal scintillator, magnified through a 10× Mitutoyo long-working distance lens yielding a pixel size of 0.65 μm. Projections were collected with an exposure time of 50 ms while the sample was rotated over 180 ° with 1.2 °·s$^{-1}$. 1500 projections were collected in 150 s. For the coarse-grained aggregate, 115 3-D microtomographic data sets were acquired over 36 hours, together forming
a 4D data set, with time as the fourth dimension. For the fine-grained aggregate, 379 data sets were acquired over 7 days. From



these 379 datasets, 19 were chosen for further detailed quantitative analysis. All acquired microtomographic data were reconstructed using the code Tomopy (Gürsoy et al., 2014) into stacks of 2160 images each, with dimensions of 2560×2560 pixels per image. Each of these image stacks contains a 3-dimensional representation of the sample mapped onto a 32-bit images, with the grey values reflecting the local absorption of x-rays (Fusseis et al., 2014a). Where the refractive indices

change in the sample, i.e. on edges, this absorption signal is locally overlain by a phase contrast signal (Cloetens et al., 1996). The time series dataset covers the entire duration of the experiment.

After the in-situ acquisition of the microtomography images, a fragment of the cup wall from the fine-grained aggregate was taken to conduct nano-scale imaging. Nanotomography was conducted using a transmission X-ray microscope (TXM) at the beamline 32-ID of the Advanced Photon Source of Argonne National Laboratory. A monochromatic beam of X-ray with an

energy of 8 keV was used. An X-ray objective lens corresponding to a Fresnel zone plate with 60nm outermost zone width was used to magnify radiographs onto a detection system assembly comprising a LuAG scintillator, a Zeiss 5X optical microscope objective lens and an Andor Neo sCMOS camera. Nanotomography yields a pixel size of ~60 nm after binning.

**2.3 Image Processing Procedures**

Zhu et al. (2016) conducted simplified analyses and measurements on 2-dimensional image slices (see Figure 2b) through the

3-D microtomography datasets of fine-grained aggregate acquired in this experiment. In this study, we present the results of a true 3-dimensional volume quantification of the microstructural changes in the sample (i.e., spatio-temporal changes in grain and pore volumes).

For the fine-grained aggregate, in each of the 19 reconstructed volumetric datasets, a sub-region that included both, the cup wall as well as the cap interior was chosen for detailed inspection. Within that sub-region, two subvolumes (see subvolume 1

and 2 from Figure 4) with a dimension of 400×400×400 voxels (260×260×260 μm$^3$) were cropped from the cup wall in all datasets. Subvolume 2 was further cropped to a volume of 247×400×400 voxels (160.55×260×260 μm$^3$) to eliminate the boundary of the cup wall.

Image segmentation is the separation and extraction of phases of interests from the 3D data sets for further analysis and quantification. A large range of segmentation algorithms exist (e.g. Kaestner et al., 2008). In global binary thresholding, images

are segmented by identifying the grey value range representing a phase and assigning all voxels within that range a single value (usually 1) while all other voxels are classified as matrix (and assigned a different single value, usually 0) (Heilbronner and Barrett, 2014). Global binarization was conducted in Avizo Fire 8 to isolate pores from solids.

At the given spatial resolution, we could not resolve the new crystals precipitated within the cup wall and only the pore space was segmented there. Pixels with grey values that fall in the range (-0.00031, -0.000077) were assigned to pore space. We

used the segmented data to quantify the change in the spatial distribution of pores during the experiment. Each subvolume (Figure 4) was further divided into smaller cubes (side lengths ~26 μm) in which the average porosity is calculated to examine where changes in porosity occurred.

In the nanotomography data, the grey value range between (2.97672×10$^{-9}$, 0.13161) was assigned to pore space. In these data, the olivine is represented by grey values between (0.30846, 1.3161). Voxels with intermediate grey values (0.13161, 0.30846)

were assigned to reaction precipitates (e.g. magnesite).





## 3 Data analysis and results

In our experiments, we have observed the development of secondary pore space during the reaction in both the coarse- and fine-grained olivine aggregate experiment (Figure 5). However, detailed examination has further revealed the differences between the formations of these pores.

### 3.1 Reaction advance in the coarse-grained olivine aggregate

In the coarse-grain aggregate experiment, the sample is not observed to be dominated by stress-generated fracturing, despite the fact that samples with larger grain sizes are generally weaker (less cohesion) compared to their fine-grained counterparts (e.g. Eberhardt et al., 1999; Singh, 1988). Within the duration of the reaction, hardly any precipitation was observed in the cup wall. The surface layer of the loose olivine grains remained free of precipitates. Both sides of the cup wall remained straight

and showed no spalling due to precipitation-caused non-uniform stretching. Instead, dissolution prevailed in the sample. The planar dissolution feature formed within 30 hours of reaction, before intense precipitation took place. Grains of olivine in the cup wall were observed to shrink in size as the reaction proceeded. The secondary pore space formed first at the center of the cup wall and grew outwards (Figure 5a). Shrinkage of the olivine grains created planar dissolution features that transected the cup wall (Figure 6). As shown by the images, the dissolution feature developed along a main plane with no obvious secondary

feature branching out.

### 3.2 Reaction advance in the fine-grained olivine aggregate

In contrast, the fine-grained aggregate experiment has shown a disintegration of the sample's cup wall with the development of fractures (as reported by Zhu et al., 2016). The post-reaction sample was fragile, contained a system of complex fractures and was essentially cohesion-less. An examination of the reconstructed 3-D images revealed the initial development of

fractures at around 68 hours after the start of the reaction. The development of the fractures exhibits a hierarchical manner similar to the description of Iyer et al. (2008) and Jamtveit et al. (2008). Figure 5b shown that the fractures first occurred at areas close to the surface and then started to propagate inwards. The fracture first developed along a single, main crack (10 ~ 15 μm) and then began to branch out. The main fracture in combination with all its branches ultimately divided the subvolume into several polygonal sub-regions. All fractures intersected with each other and formed a systematic pattern in which most of

the divided regions maintained the shape of regular polygons (Figure 7). Most of the later fractures emerged in directions which are at high angles (70 ~ 90 °) to previously existing fractures, leading to sub-regions that had a distinct shape with four sides similar to the description of Bohn et al. (2005) for fractures that developed successively. The pattern of the network of fractures are also similar to those developed during the mud desiccation (Plummer and Gostin, 1981).

The grey value histograms of 4D microtomography data evolve systematically during in-situ experiments, which can be

utilized in their evaluation (Fusseis et al., 2012). Systematic analysis of the histograms of the grey value distribution revealed the progression of reaction during the experiment. In our data, we observed that the histograms became flatter and wider over the duration of the experiment, with an increase in the number of the darkest and brightest voxels at the expense of the voxels with intermediate grey values (Figure 8). These systematic changes in the absorption behaviour can only be caused by the sample reacting and indeed reflect the dissolution of olivine, the generation of pore space and the precipitation of reaction

products, in addition to phase contrast around newly generated edges in the sample. The best fit curves to the histograms evolved systematically during the reaction and intersected in a relatively narrow grey value range (-0.000077, -0.000055) (Figure 8). Voxels with grey values darker than -0.000077 correspond to fluid-filled pores, whose volume proportion increases throughout the reaction process.



In the cup wall, we observed patches that gradually cemented with fine-grained reaction products (Figure 7), though individual precipitates could not be identified. The texture of these patches became smoother with time, consistent with the disappearance of pore space. The nanotomographic data from the cup wall confirm this observation. Nanotomography images clearly show the existence of precipitates in between olivine grains (Figure 3 from Zhu et al., 2016). In the nanotomographic data, the x-ray attenuation coefficients of the precipitates corresponded to a grey value range from 0.13161 to 0.30846, distinctly different from those of olivine (> 0.30846) and pores (< 0.13161). EDS analysis showed a significant carbon peak within the precipitates, which strongly suggests the growth of magnesite (Zhu et al., 2016).

Growth of precipitates within the cup wall led to an expansion of the volume. To quantify the volume expansion, we traced several gains in the center of the cup wall. Distances between these grains were measured at different times. Our results indicate an average expansion of ~9.1 μm over a distance of ~260 μm from 7 hours to 125.9 hours after the start of the experiment, which corresponds to a volume expansion of 2.8% ~ 4.7%. However, little to no expansion was observed at the edge of the sample.

### 3.3 Dissolution Feature vs fracturing

The dissolution and fracturing are both observed to generate secondary pore space during the experiments, but the dissolution feature differs from the reaction-induced fractures in many ways:

- Firstly, the dissolution feature is a single, planar feature in 3-D, while the fractures observed in the fine-grained aggregate formed a network of intersecting cracks. Figure 9 shows the morphology of the dissolution plane and the fracture network. It's obvious that the fractures intersected with each other and formed a complex wedge shape network with the vertex pointing towards the sample's interior. The dissolution feature mainly developed alone a plane and show less intersection with other features.
- Secondly, the sample dominated by dissolution features shows clear evidence for shrinkage of larger grains and disappearance of smaller grains at the interior of the aggregate (Figure 5a). The fine-grained aggregate exhibit patches that develop during the reaction which are evidences for the reaction product precipitation.
- Thirdly, the development of micron-scale dissolution is simultaneous, with the shrinkage of grains occurring both at the surface and the interior of the cup wall (Figure 6 and 9b). In contrast, development of the reaction-induced fracturing is successive with most fractures occurred first at the surface and migrated towards the interior of the cup wall. This caused the observed wedge shape of the fracture network (Figure 9a). Development of the fractures also exhibits hierarchical sequence with main fractures appeared first. The secondary fractures branch out from the main fracture and divide the sample into smaller patches (Figure 7).
- What's more, no precipitates were observed along planar dissolution features, the cup wall remains straight throughout the experiment. But for the fine-grained aggregate, the cup wall shows clear spalling which is a sign of none-uniform expansion that links to precipitation in this experiment.

### 3.4 Individual olivine grain

The reaction affected not only the aggregate but also the individual olivine grains. Both the dissolution feature and reaction-induced fracturing are observed at grain scale. Figure 10 shows a series of image slices through a nanotomography dataset, moving through an olivine grain in the cup wall. The grains clearly exhibit channels (etch pits) in the reaction zone. In video S1 (supplementary materials) it can be seen that these channels penetrate into, and even through olivine grains. The tubular shape and the depth of penetration indicate that they are 'worm hole' features, likely resulting from dissolution. The





shape and the width of these channels vary, with wider inner channel diameters below the surface suggesting more extensive dissolution at depth.

The fracturing is also observed on individual grains at nano-meter scale. Figure 11 shows nanotomographic evidence for hierarchical fracturing within olivine grains. The secondary fractures developed from the primary fracture and formed in a direction that is perpendicular to the earlier ones.

In the reconstructed microtomographic images of the loose grains from the fine-grained aggregate cup, we observed the precipitation of secondary minerals on the surface of the olivine grains inside the cup (Figure 12). On the basis of their rhombohedral shapes we identified these as magnesite crystals. Other minerals (e.g. serpentine, brucite, etc., see reaction 1~4)
were likely also present in the sample, but could not be isolated at the given image resolution and absorption contrast. Measurement of the magnesite circumference revealed continuous growth during the experiment (see Figure 5 in Zhu et al., 2016). The first magnesite crystals emerged after 48 hours, and grains kept nucleating and growing after that. Growth continued until the experiment was aborted and no deceleration could be observed at any point, which indicates that the sample continued reacting. We determined a growth rate for the grain perimeter of $0.772\ \mu m \cdot hour^{-1}$, which, by assuming cubic shape of the
crystal yields an equivalent growth rate of $7.18 \times 10^{-3}\ \mu m^3 \cdot hour^{-1}$. We used a density of $3.01\ g \cdot cm^{-3}$ and molar weight of $84.314$ $g \cdot mol^{-1}$ for magnesite ($MgCO_3$) in our calculation. Assuming a specific reaction surface of $50 \times 50\ \mu m^2$, this gave a magnesite growth rate of $2.85 \times 10^{-15}\ mol \cdot cm^{-2} \cdot s^{-1}$. This is in general in agreement with the calculation of Saldi et al. (2009).

The volume change of a loose olivine grain inside the cup was calculated to quantify the competing effect of dissolution and precipitation during olivine carbonation. Individual grains were labeled from the segmented data, and their volumes
determined. Figure 12 plots the volume change of a single grain selected from the loose grains inside the fine-grained cup over 13 successive microtomographic datasets (covering 146 hours). Magnesite overgrowth causes a significant roughening of the olivine grain surface. While the volume of the individual magnesite grain steadily increased throughout the reaction (Zhu et al., 2016), the total volume of the grain (olivine plus precipitates) fluctuates from time to time, which reflects variable rates both in the precipitation of magnesite as well as in the dissolution of olivine. A large drop in grain volume occurred at around
38 hours, which is consistent with a period of vigorous olivine dissolution. The largest continuous grain volume increase took place between 40 to 70 hours, caused by the precipitation of magnesites. At ~70 hours, the grain volume again decreased considerably, indicating that dissolution became dominate once more. This second dissolution episode coincides with the appearance of reaction-induced fractures in the aggregate wall at ~68 hours, suggesting a positive feedback process.

## 4 Discussion

We claim that our experimental observations indicate the activity of two different mechanisms that both create fluid pathways effectively. These are dissolution-dominated fluid pathway generation at micro-meter scale in the case of the coarse-grained aggregate and at nano-meter scale in the case of fine-grained aggregate, and reaction-induced fracturing in the case of the fine-grained aggregate. We detail our interpretation in the following sections.

### 4.1 Dissolution and etch-pitting

Dissolution and etch-pitting are important mechanisms that affect the grains' surface morphology and the permeability of the sample (e.g. King et al., 2010; Røyne and Jamtveit, 2015). The nanotomographic observation, that etch-pitting incurs extensive dissolution beneath grain surfaces, was also documented by Lisabeth et al. (2017a, 2017b) during the carbonation of dunites. Peuble et al. (2018) also observed nano-meter scale veinlets forming oblique to sub-vertical channels in partially-carbonated





olivine grains during percolation experiment. The hollowing out of olivines seems especially important in areas where the
grain boundary porosity is decreasing due to the precipitation of secondary minerals. There, subsurface dissolution channels
in olivine grains preserve important fluid pathways and maintain the reaction. This supports the hypothesis of Andreani et al.
(2009) that the permeability can be maintained by the preferential dissolution even in cases where the overall porosity is
decreasing. Apart from providing access for fluids, subsurface dissolution features also make the grain more susceptible to
fracturing and thereby promote the generation of fracturing observed in the experiment.

The permeability of these dissolution channels was calculated using Hagen–Poiseuille's Law (Poiseuille, 1844) and Darcy's
Law (Darcy, 1856). Hagen–Poiseuille's Law allows the calculation of the fluid flow inside a tube in dependency of the pressure
gradient:

$$Q = \frac{-\pi r^4}{8\eta}\frac{\Delta P}{\Delta L},\tag{1}$$

where Q is the volume flow/rate of discharge, r is the radius of the tube, $\eta$ is the viscosity of the fluid, $\Delta P$ is the pressure
difference over the flow length $\Delta L$. Darcy's Law stated that the rate of discharge is proportional to the viscosity of the fluid
and the pressure drop over a given distance, i.e., for a tube with a radius in r:

$$Q = \frac{-k\pi r^2}{\eta}\frac{\Delta P}{\Delta L},\tag{2}$$

where $k$ is the permeability of the tube. Combining the two equations, $k$ can then be derived as:

$$k = \frac{r^2}{8},\tag{3}$$

In this experiment, $r$ is ~6 pixels (360 nm). This yields a permeability of ~$1.6\times10^{-14}$ m$^2$, which is high enough to transmit
reacting fluids through these channels. Considering the density of 'worm hole' features over the grain, the permeability
contributed by the channels could be reduced by two orders of magnitude but is still able to transmit fluid.

These intragranular channels contrast the transgranular fractures. But the development of these etch-pitting dissolution
channels also provide fluid path for the reaction and allow a more extensive degree of alteration of the grain. They weaken the
grains, make them more susceptible to disintegration and provide nucleation sites of the later fracturing. While sparse in our
data, we think that over geological time scales the contribution by dissolution channel to bulk permeability and the advance of
the reaction would be significant. However, on the time scale of our experiments, these features alone are insufficient to explain
the observed self-sustainability of the reaction considering the scale and density of the dissolution channels and we argue that
the main contribution must come from volume mismatch cracking in our laboratory study.

### 4.2 Reaction-induced fracturing

Olivine carbonation could produce up to 44% increase in solid molar volume assuming the reaction can proceed to completion.
If such a volume increase takes place, the crystallization pressure generated could be high enough to fracture the host rock
(Kelemen et al., 2013; Kelemen and Hirth, 2012). However, experimental studies on olivine carbonation show no evidence of
high crystallization forces (van Noort et al., 2017) but rather suggest that precipitation causes the pore space to fill up and halt
the reaction before the crystallization induced pressure reaches the critical value needed to generate fracture (Hövelmann et
al., 2012; van Noort et al., 2017). Our quantitative estimates indicate that in these experiments, crystallization pressure can




lead to maximum ~5% volume expansion. This is not enough to break the host rock, as shown in salt crystallization experiments (Scherer, 2004). Indeed, the nanotomography data show only dissolution features such as etch pits and worm holes, with no evidence of cracks in olivine grains surrounded by precipitates (Figure 10). The lack of evidence for crystallization pressure-induced cracking is consistent with other experimental studies (e.g. Hövelmann et al., 2012).

We interpret our observations from the fine-grained aggregate (analyzed in less detail in Zhu et al., 2016) as evidence for reaction-induced fracturing, and the observed fracture patterns to form analogous to shrinkage cracks (e.g., desiccation cracks, Edelman, 1973; Plummer and Gostin, 1981). In Zhu et al., (2016)'s model, the loose olivine grains inside the cup act as precipitate traps that keep the surface of the cup wall relatively free of precipitation. In the interior of the cup wall though, away from the precipitate traps, the crystallization pressure builds up and causes expansion. While the crystallization pressure is too low to cause shear fracturing of the cup, the expanding cup wall interior stretches the surface of the cup wall and causes it to fail in tension and tear. This is facilitated by the near-constant surface area (which decreases slightly as a result of dissolution). In analogy with desiccation cracks, the resulting fractures form characteristic and systematic polygonal patterns: The first set of fractures intersect at right angles, and all subsequent fractures divide the sample into smaller polygonal domains with increasing intersection angles. Since the fracture pattern develops successively rather than simultaneously, the higher-order fractures form in a different stress geometry and as a result migrate perpendicular to the surfaces generated by the previous fracturing event.

To evaluate the potential of surface stretching as a fracture generating mechanism, we estimated the stress that could be produced by the volume-mismatch in the cup wall. We did so by identifying and tracking grains whose spatial coordinates (x, y, z) changed continually as the sample expanded. Measurements of the distance between grains revealed an axial elongation of 2.78~4.71% in ~120 hours. This would translate into an axial strain of ~0.03 of the outer layer in order to compensate the volume mismatch. The elastic moduli and strengths of the synthesized porous olivine aggregates are similar to weak sandstones. Using a Young's modulus of ~10 GPa yields extensional stresses generated due to the expansion of ~300 MPa, easily exceeding the tensile strength of the sample (~10MPa). Interestingly, our estimate of extensional stress generated by the volume mismatch is of the same magnitude to the stress from crystallization pressurization (e.g. Kelemen et al., 2013). For natural peridotite, Young's moduli range from 108 to 194 GPa (Christensen, 1966), tensile strength is 50 to 290 MPa and spall strength is ~58 MPa (He et al., 1996). An equivalent volume expansion of ~10% in nature could lead to a stress of 3.24 GPa. In both cases, the stress is more than sufficient to fracture the material. However, these are simple estimations of stress and strain made with basic assumptions and local conditions. Considering the extent of the local carbonation reaction and how the expansion in the center is affecting straining of the outer layers, the estimated stress can be considerably lower but should still in a range that is sufficient to break the material.

To generate the expansion cracks via surface stretching, the volume mismatch must be substantial, which requires to keep the near surface region free of precipitates. Zhu et al. (2016) suggested that the loose olivine grains inside the sample cup worked as precipitate traps/attractors in this experiment. Because the rate of crystal growth decreases drastically as the curvature of the substrate increases (García et al., 2013; Ziese et al., 2013), large grains in general are preferred sites for precipitation of new crystals. With a size contrast of ~2 orders magnitude, the loose olivine grains (100-500 μm) in the immediate vicinity of the inner cup surface fulfilled the function of precipitate traps and thereby kept precipitation level at the surface of the olivine cup wall low.

We tested the idea of volume mismatch cracking by conducting the coarse-grained (80-100 μm) aggregate experiment. Now the size contrast between the grains forming the aggregate in the cup wall and the loose grains inside the cup was significantly reduced, and we expected less efficient precipitate trapping, and consequently little to no reaction-induced cracks. The



experimental results support this idea. The only planar features observed in new microtomography experiment are planar dissolution features (Figure 6).

A detailed examination of the dissolution channel shown in Fig. 6 revealed no evidence of precipitates there. This places doubt
on the crystallization pressure being responsible for fracturing during olivine carbonation. If not, what could be an alternative explanation for the observed fracturing? To further address this question, we examined the porosity evolution and distribution in the 3-D tomographic datasets.

Despite the histogram analysis revealed a bulk increase in the porosity of the cup wall during the experiment, the distribution of these newly generated pores was inhomogeneous in the sample (Figure 13). This perturbed the initially homogeneous
porosity distribution. This effect became particularly apparent in subvolume 2, which started to exhibit higher porosity where it was closest to the outer surface of the sample after 68 hours (see Video S2, S3, S4 for details). The difference in porosity distribution within the sample became more pronounced as the reaction proceeded, with porosity in the outer surface of the cup wall increasing while the porosity inside the cup decreased. In our interpretation, this change in the pore volume reflects a contrast in the precipitation rate where precipitation proceeded more slowly in the outer part compared to the inner part of
the sample wall, leading to different rates of expansion and the generation of tensile stresses.

The 'expansion cracks via stretching' mechanism can explain the observed microstructure evolution in two subvolumes (Figure 4). Since subvolume 2 is located at the periphery of the cup, it would be fractured before subvolume 1 which locates in the center of the cup wall. This predict from the 'expansion cracks via stretching' is consistent with the observed distribution of pore space that most porous area locates close to the periphery. This mechanism also explains why the fractures tend to develop
perpendicular to the reaction surfaces. Similar models relating the reaction generated stress and fractures has also been used to explain other exfoliation cracks (e.g. Blackwelder, 1925).

In summary, two fundamental observations from our experiments are inconsistent with the "internal cracks generated by crystallization pressure" mechanism and form strong arguments against the existence of any significant crystallization forces (Figure 10, 13). Firstly, if the cracks were generated directly by crystallization pressure, we should expect them to initiate in
region with intense precipitation and porosity reduction. However, in the outer layer of our sample where the fractures are observed, no precipitates formed prior to the fracturing. Secondly, on the nanoscale, the channels formed by dissolution show hollow and smooth inner surfaces and no precipitation of magnesite (or any other minerals) which shows evidences of low crystallization pressure.

Our detailed analyses provide quantitative support to the "surface cracking via volume mismatch" model first proposed by
Zhu et al. (2016). Previous experimental studies on olivine carbonation show that the crystallization force is low (van Noort et al., 2017), suggesting that breaking host rocks by crystallization pressure as in salt crystallization is unlikely which is in contrast with fracture networks that are commonly observed in naturally occurred serpentinized and carbonated peridotites (Iyer et al., 2008; Macdonald and Fyfe, 1985). We present a resolution to this conundrum by documenting a process that allows fracturing without a high crystallization force.

**4.3 Coupled-mechanisms of dissolution and precipitation-driven fracturing**

The findings of this study can be summarized in a mechanism that couples dissolution and precipitation during olivine carbonation. If dissolution and precipitation are heterogeneously distributed in a rock, non-uniform volume expansion can cause breaking of the host rock via surface stretching. In nature, heterogeneity in the porosity and permeability of a rock formation is common, which may cause non-uniform concentration of reaction and distribution of precipitation (Wells et al.,





2017). As shown in our study, the resulting volume mismatch could lead to expansion fractures. The fractures provide new fluid pathways and expose fresh reactive surfaces to sustain the carbonation. In a long-term, fluid pathways may be provided by "worm-hole" etch pitting. Dissolution channels could deteriorate rock strength over longer time scales (Figure 14).

In general, several different mechanisms seem to facilitate olivine alteration and contribute to sustaining it. On relatively short time scales, rapid reaction-induced tensile fracturing could be the dominating mechanism that maintains the reaction, whereas
on a longer timescale, dissolution and the formation of channel-like structures may dominate.

## 5 Conclusions

Using synchrotron-based micro- and nano-tomography, we documented and quantified the reaction progress during olivine-fluid interaction on the micron scale. This allowed us to identify mechanism that sustain the reaction despite its large positive volume change.

In summary, our experiment results suggest:

- The reaction-induced fracturing observed in our experiments results from non-uniform volume expansion. Tensile stresses arise from heterogeneous precipitation and the resulting contrast in the expansion.
- Even though the dissolution cannot be used alone to explain the sustainability of the experimental-time-scale olivine carbonation, it provides evidence that dissolution etch-pits can provide fluid path and fresh reaction surface for the
reaction to proceed. This helps in explaining the naturally occurring complete alteration of peridotite, as the time scale for natural carbonation ranges from thousands to million years. Even if the dissolution channelizing would only allow slow fluid flow, it could still induce significant alteration given time.
- The coupled-mechanism of dissolution and reaction-induced fracturing accounts for maintaining the reaction processes during olivine carbonation. It explains on different time and space scale about the formation of observed
natural outcrops of completely carbonated peridotite.
- The results from our experimental study also provide new insights into the application of $CO_2$ mineral sequestration.

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



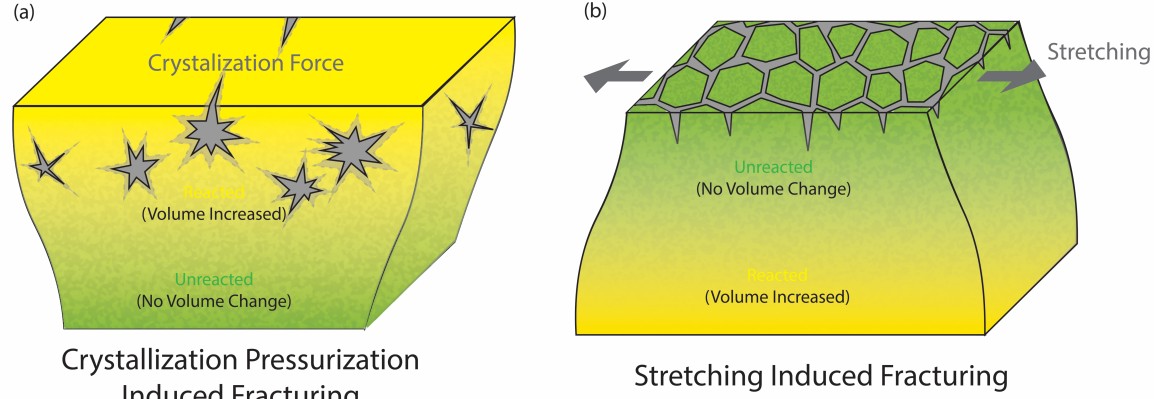

**Figure 1: Illustration of the mechanisms of reaction-induced fracturing during olivine-fluid interaction. a) The crystallization**
**pressurization described the development of fractures caused by forces exerted on the surroundings due to growth of precipitates.**
**Salt crystallization (Scherer, 2004) is a typical example of the crystallization pressure induced fracturing. The fractures first appear**
**at areas where precipitation is most concentrated and propagate outwards. b) The surface cracking model describes the development**
**of fractures as a result of a contrast in expansion which causes stretching at the surface. A difference in the precipitation rate between**
**the periphery and interior of the sample causes them to expand in different rates with the inside expanding faster than the outside.**
**This builds up the tensile stress at the surface that fractures the sample and leads to a development of a polygonal fracture network.**
**The fractures propagate from the surface towards the inside.**

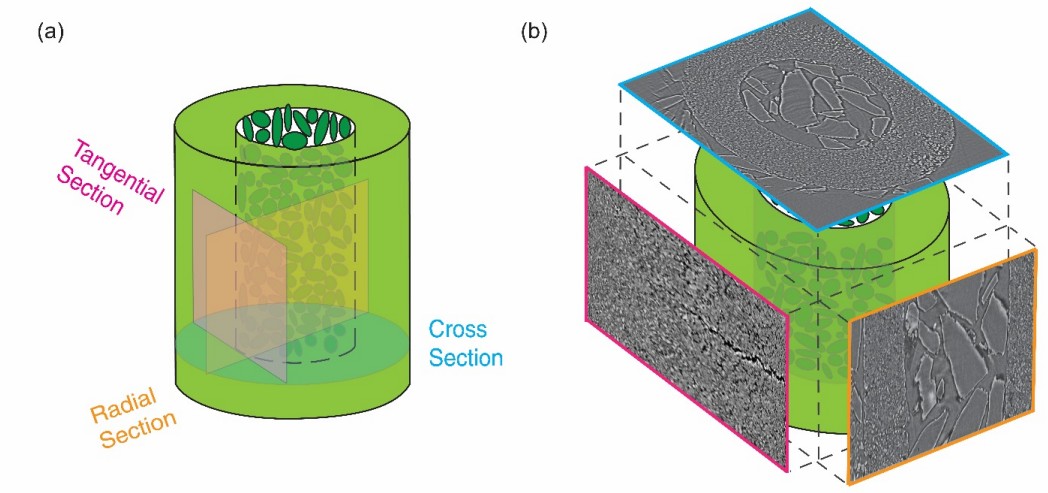

**Figure 2: Illustration of sample configuration. a) The sample is composed a sintered olivine aggregate cup with loose olivine sand**
**fillings (grain size 100-500 µm). 2D examinations of the sample are conducted on the tangential (pink), radial (orange) and cross**
**(blue) section of the sample. b) Example of the tomographic images that was used in the 2D examination of Zhu et al. (2016).**



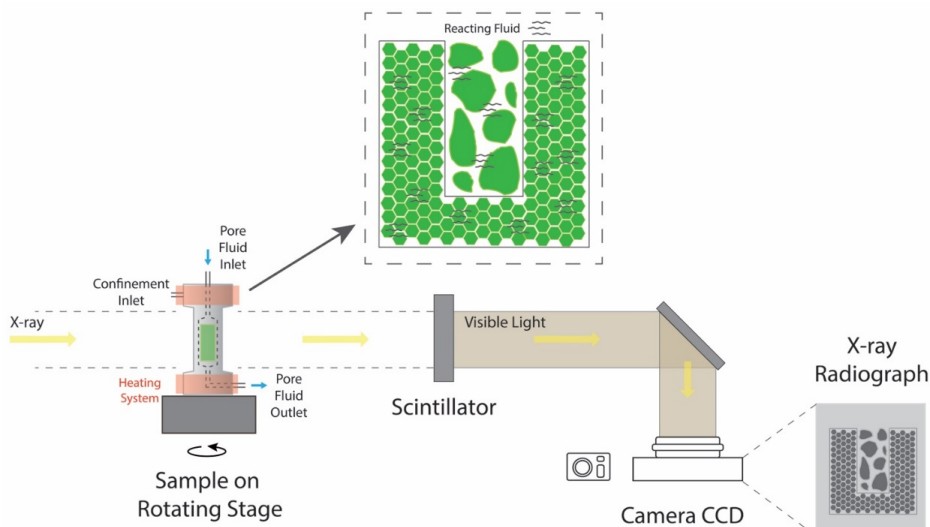

**Figure 3: Experimental setup for dynamic microtomography. Inside the x-ray transparent pressure cell, the confining pressure, pore pressure and temperature can be controlled independently. The synchrotron radiation imaging records radiographs of the sample at in-situ conditions with ongoing reaction at different angular positions with the sample being rotated.**


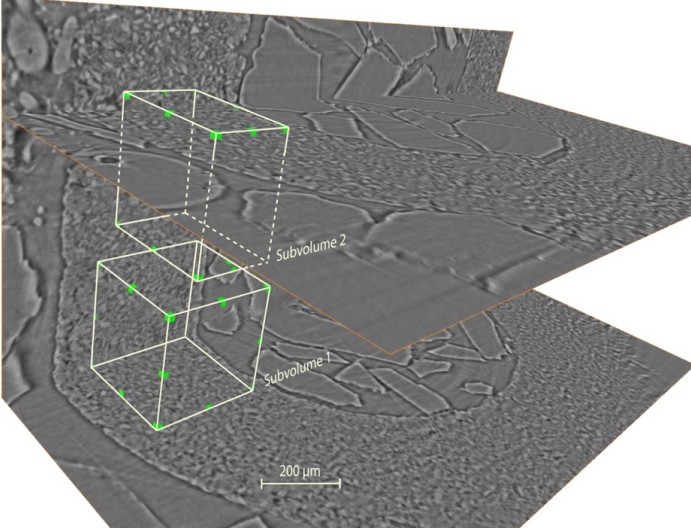

**Figure 4: The sub-region of the fine-grained aggregate sample examined in detail. Positions of the two subvolumes in the cup wall. Subvolume 1 (bottom box) is located at the center of the cup wall. Subvolume 2 (top box) is located adjacent to the outer rim of the wall.**





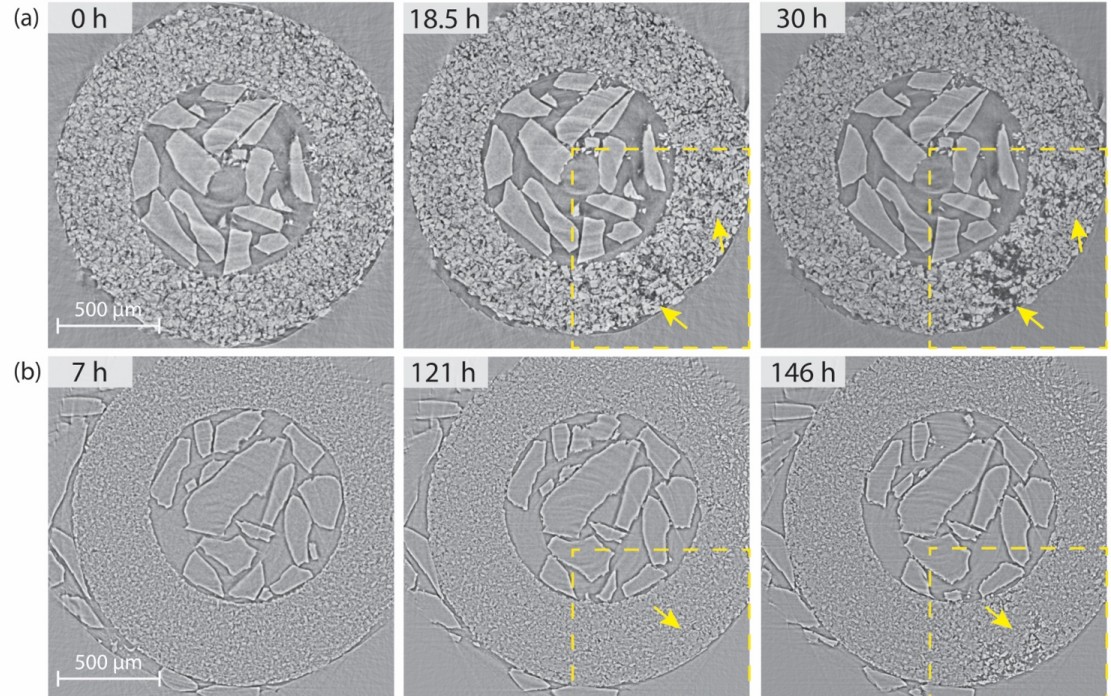

**Figure 5: Reconstructed images showing horizontal cross section view of the samples. a) Coarse-grained aggregates exhibit dissolution features in which grain of olivine shrank and formed secondary pore spaces. Most of the large pore spaces concentrate at the interior of the cup wall (highlighted by yellow dash lines). b) Fine grained aggregates show that the fractures first developed at the surface of the cup and propagated from the outer rim into the cup wall (highlighted by yellow dash lines). Larger pore space distributes mainly near the rim.**



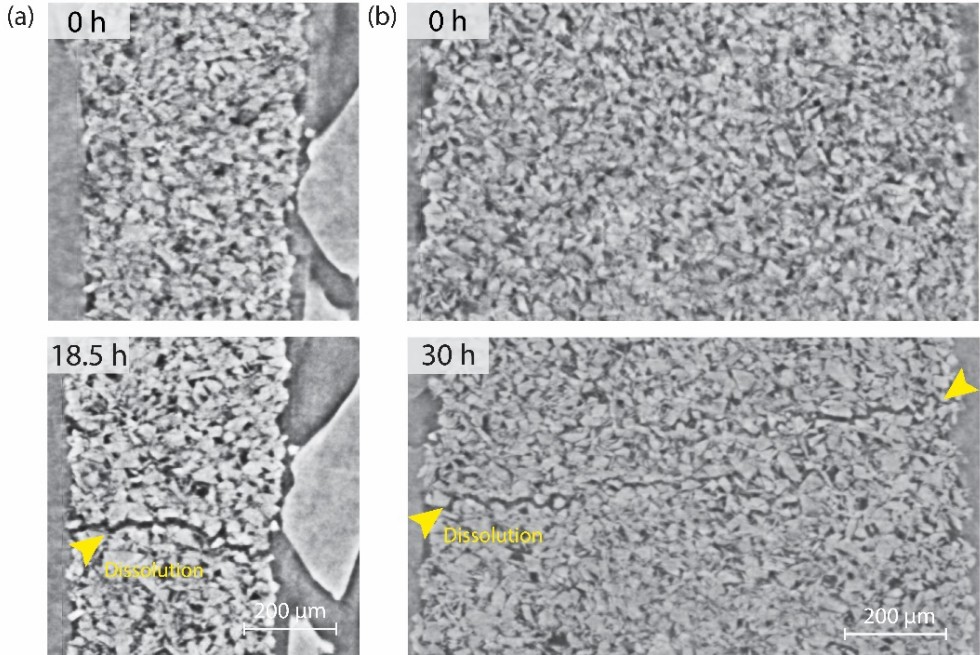


**Figure 6: a) Radial section images of the coarse-grained aggregate. Evolution of the microstructure of a coarse-grained olivine cup during carbonation reaction. Dissolution features started to develop after ~10 hours. The time series images show shrinking olivine grains around these dissolution features. The edge of the cup wall remains straight with no sign of precipitation. b) Tangential section images of the coarse-grained aggregate. The dissolution features (yellow arrows) appear smooth and planar in the radial section. No**
**evidence of expansion cracks was found.**




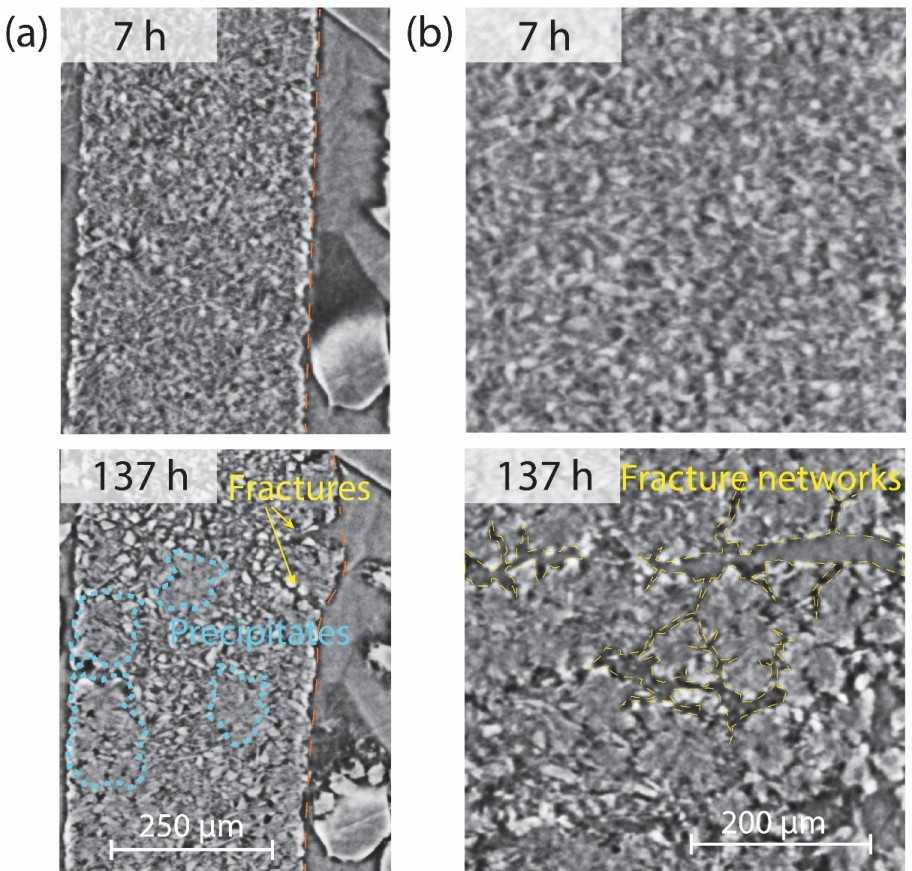

**Figure 7: a) Radial section images of the fine-grained aggregate. Cemented patches (outlined in blue) grow within the cup wall as precipitation progresses. The time stamps indicate the hours passed since the reaction started. Slices are from the exact same location at different time. Arrows point to the reaction-induced cracks. Edge of the cup wall becomes curved (outlined in orange). b) Tangential section images of the fine-grained aggregate. The surface of the expansion fractures observed in the fine-grained experiment is sharp and jaggy, branches of secondary fracturing occur at high angles (70 ~ 90 °) to the primary fracture. The fracture networks divide the sample into smaller polygonal patches. Part of the fracture networks is outlined by yellow dash lines.**






**Figure 8: a) Best-fits for the grey value distribution histograms of the sample at different stages of the reaction. Different colors**
**represent time lapses as shown. Pores, olivine, and precipitates are identified based on their grey value ranges. Higher values**
**correspond to lighter grey (solids). The more negative a value is, the darker the grey color becomes (e.g., pores are black). b)**
**Segmentation of pore space in subvolume 1 (at 146 hours). In the 2D cross-sectional images (marked as planes x, y, z), the red areas**
**represent segmented pore spaces.**



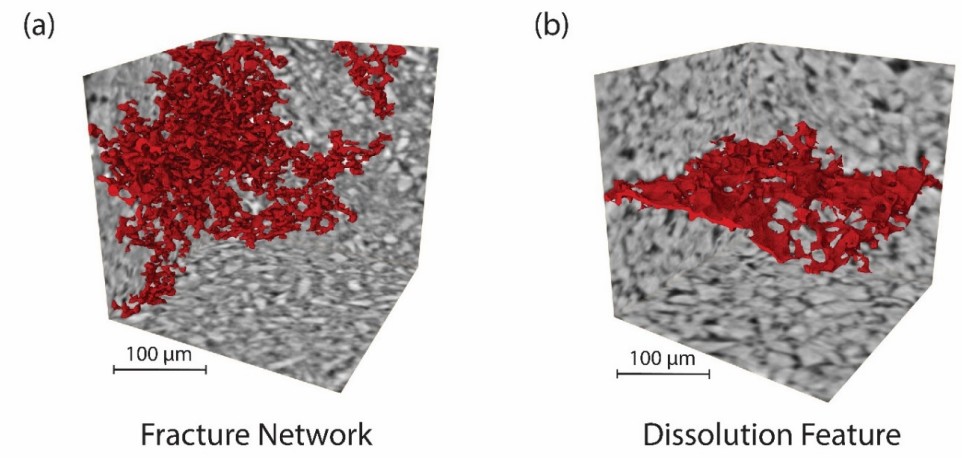


Figure 9: 3D volume of a) the fracture network in the fine-grained aggregate and b) the dissolution feature in the coarse grain aggregate. The fracture network shows a complex of fractures propagating from the surface towards the interior. The dissolution feature appear to be planar without the formation of complex network. Both displayed volume are 260×260×260 µm³ in size.

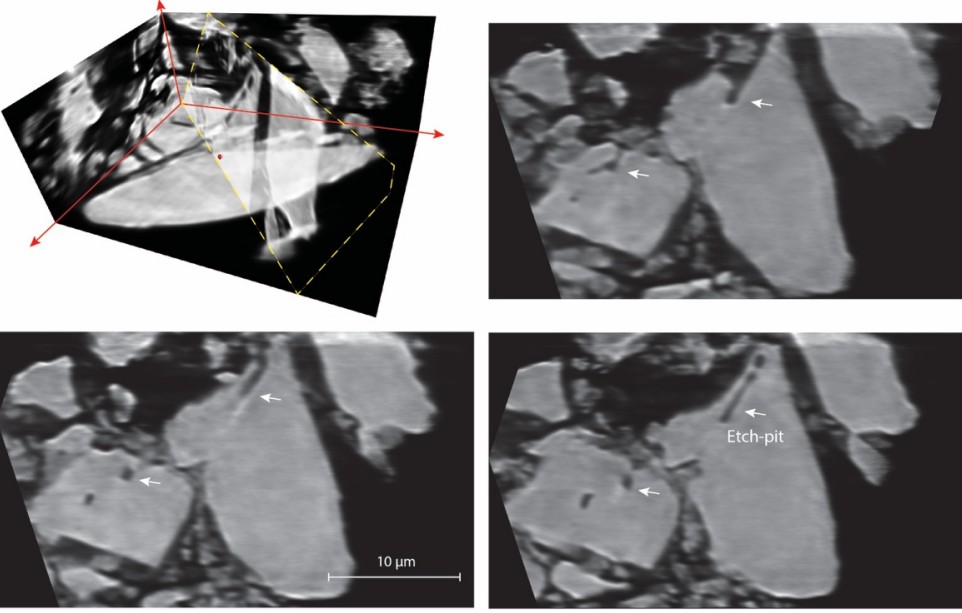

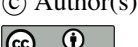


**Figure 10: Reconstructed images from the nanotomography data demonstrate the existence of etch-pits and dissolution channels (white arrows) formed in the olivine grain. The precipitates (darker grey) partially fill the pore space (black) between olivine grains (lighter grey). The yellow dash line marks the orientation of the cross-sections. The vertical distance between each 2D cross-section is ~600 nm. Reacting fluid causes a preferential dissolution of the grain which develops small channels that dig into the grain. These features provide a fluid path and eventually break grains, exposing new reactive surfaces.**

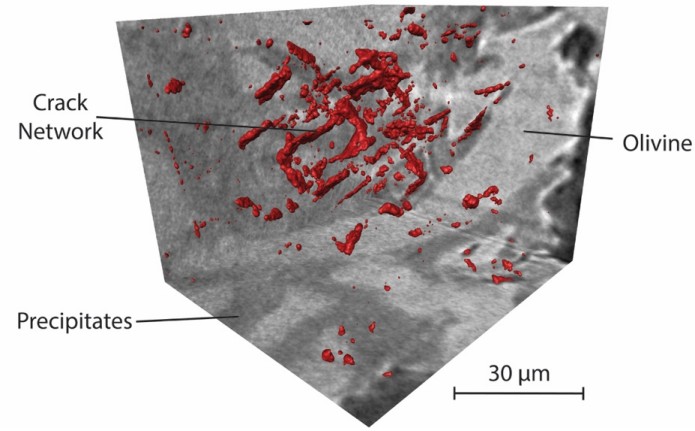

**Figure 11: Network of microcracks (red) in the reaction olivine cup wall shows a polygonal pattern. The dissolution channels make the grain (light grey) more susceptible and a stress concentration is also likely to occur around the etch pits (Plümper et al., 2012).**

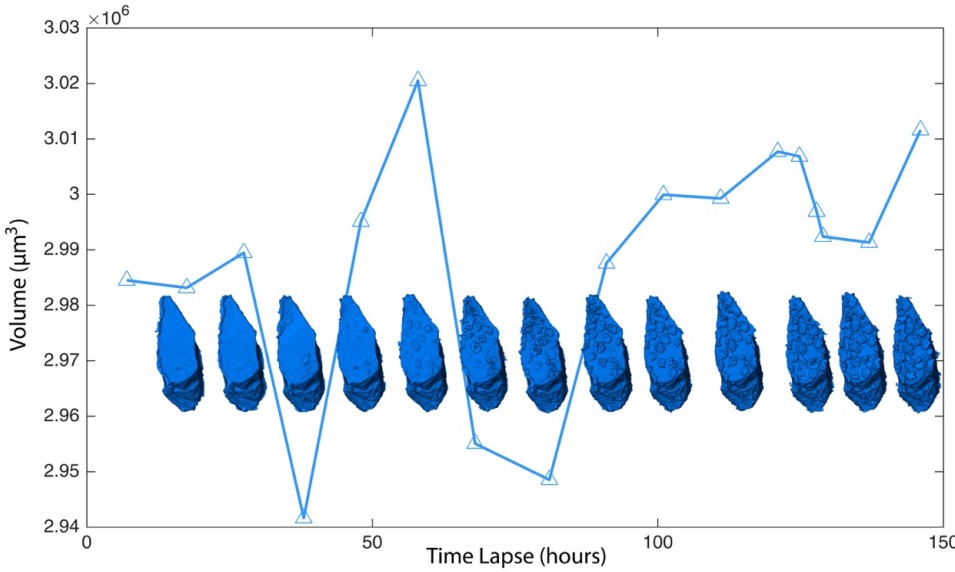



**Figure 12: Volume change of an individual olivine grain (total volume of olivine and precipitates) during carbonation reaction. The precipitates are idiomorphic and referred to as magnesites (see Zhu et al., 2016). This observed volume change results from a combination of the dissolution of olivine and precipitation of magnesites. Fluctuations of the grain volume manifest the altering dominance of dissolution versus precipitation.**

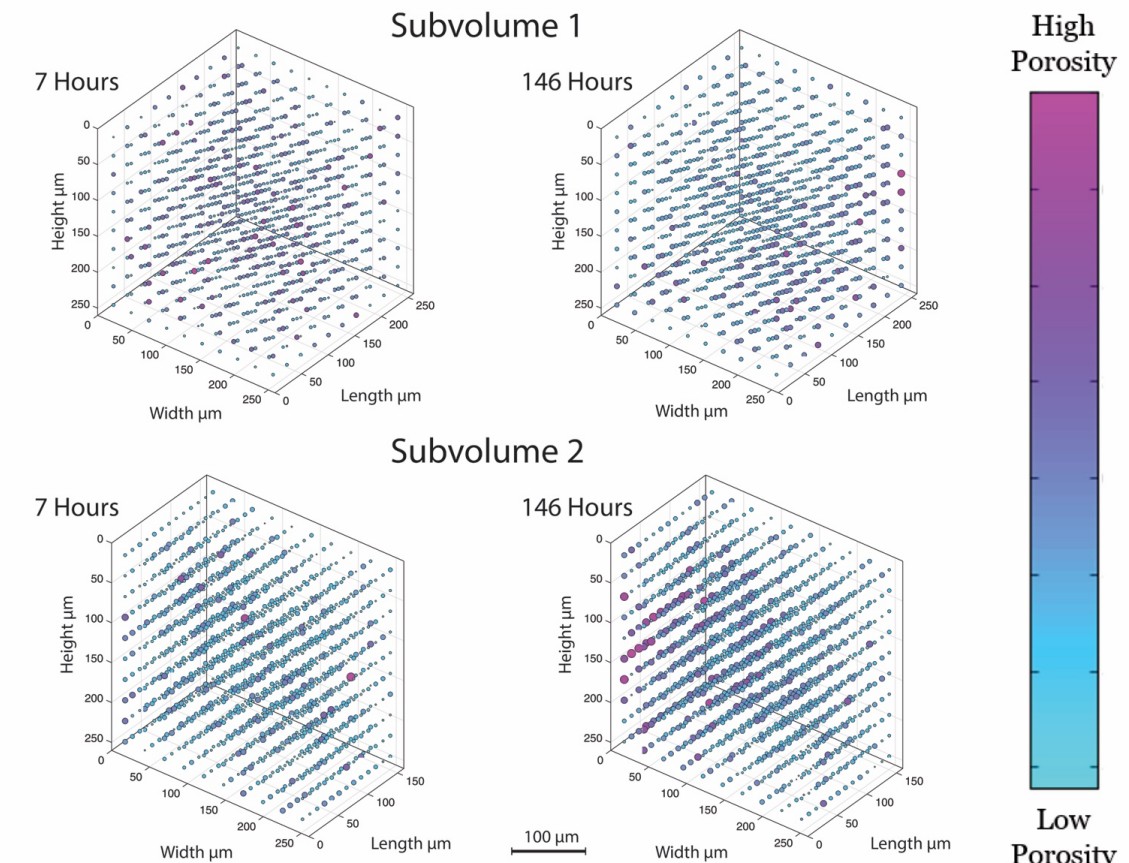

**Figure 13: Pores distribution within the sample. For subvolume 1 which contains more inner part of the sample, porosity is relatively homogenous among the volume. For subvolume 2, as it contains more outer part of the sample, a concentration of high porosity can be found in the outer edge compare to the inner edge of the subvolume. This contrast in porosity also reflects a non-uniform precipitation which generates stress that fractures the rock.**





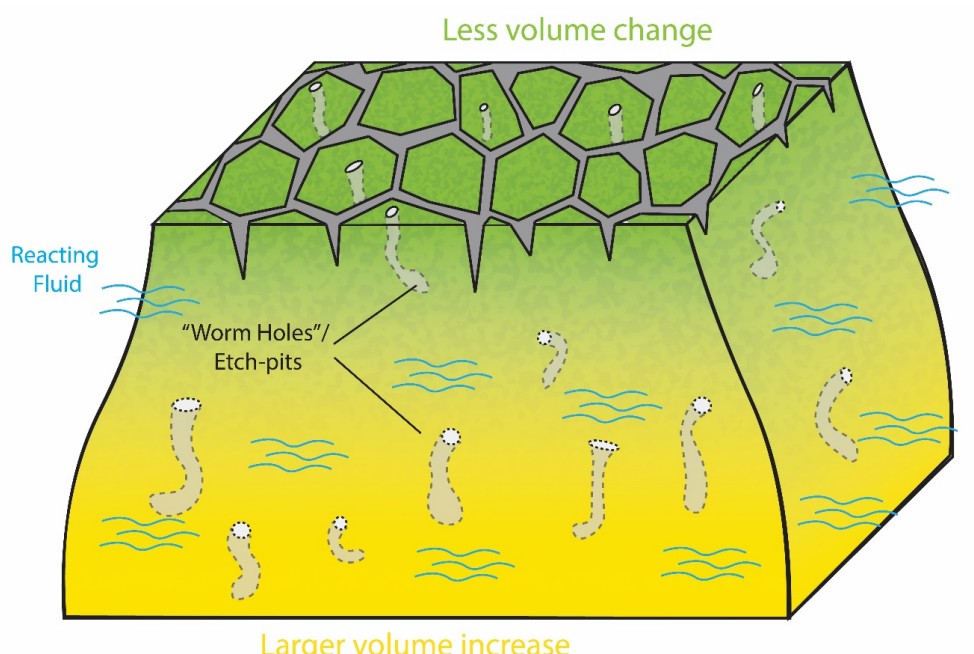

**Figure 14: Illustration of porosity generation mechanisms during the olivine carbonation reactions. A combined mechanism of surface cracking and the dissolution channelization plays an important role in the porosity generation. Heterogeneity in the micro-structure of the material would cause none-uniform distribution of precipitation. This would lead to the generation of surface cracking via volume mismatch and generate secondary porosity. Dissolution also produces pore spaces and fluid pathways through etch-pitting channelization which makes the grains susceptible for the cracking on a longer time scale.**