# Peer review of "Generating porosity during olivine carbonation via dissolution channels and expansion cracks"

_Solid Earth, 2018_

## Referee Comment (RC1) · Anonymous Referee #1 · 17 May 2018

**1/ Introduction**

The manuscript by Xing et al. entitled "**Generating porosity during olivine carbonation via dissolution channels and expansion cracks**" reports on a very nice piece of experimental work on the in-situ hydrothermal carbonation of olivine aggregates. Careful attention is paid to the real-time development of microstructures to unravel reaction-induced porosity changes and fracturing. Indeed coupling and feedbacks between dissolution/crystallization and generation of new fluid pathways within mineral aggregates (synthetic rock) is still poorly known although highly relevant to metamorphic and alteration reactions which involve aqueous and carbonate fluids. The study basically confirms a reaction-transport-deformation model that has been proposed by the same authors in 2016 based on a very similar experiment using the same characterization technique. The difference with the present study mostly relies on the use of a different mineral grain size. The high similarity between the two studies makes sometimes difficult to distinguish between data that have been collected here and in the previous study. Naming samples like LGC (larger grain cup) and SGC (smaller grain cup) would potentially help.

**2/ General comments**

- **2.1 About Reaction progress**

    Estimate of the overall reaction progress in the cup is an important piece of information. In a system that is prone to porosity clogging due to volume expansion of the solid phases, it is expected that the nature, density and geometry of fluid pathways will change with reaction progress. Basically, are the features described in this study relevant to peridotites that are at the beginning of the carbonation process (< 10%) or do they apply to extensively carbonated systems?

    Basically, if one considers a solid volume (Vs) expansion of x, then the volume expansion of the solid matrix ($\Delta$Vs) is a function of reaction progress (R): $\Delta$Vs = V°s*R*x. Let consider the end-member case where the overall sample volume (Vr) is constant and that the expansion of the solid matrix is only compensated by porosity shrinking. Then, porosity will vanish when $\Delta$Vs = Vr(p°) where p° is the initial porosity. Finally, we end up with R = p°/(1-p°)x. In the SGC cup, p°= 0.1. Assuming x = 0.4 (40% solid volume expansion) then R = 0.3. We see that for reaction progress above 30% in the cup, porosity could have potentially vanished at constant sample volume.

    Obviously, the constant volume assumption does not hold in the present case but this simple calculation shows why knowledge of the reaction progress is conceptually important.

    Following this idea, the knowledge of the three parameters, sample overall volume expansion, average porosity and reaction progress when the experiment is terminated would be very useful.

    The knowledge of the overall reaction progress is also important if the experiment is run in a close system (technical point to be clarified) the source of $CO_2$ will be limited. In a forsterite sample with 10% porosity, all the $CO_2$ initially supplied as $NaHCO_3$ will be consumed after 20% reaction progress.

- **2.2/ About the Model**

    I see an alternative model to the stretching-induced fracturing model. The inner cup contains loose grains and the porosity is the highest there. Accordingly, most of the solution is located there, solution which, furthermore, can be partially renewed if the system is not fully close (inlet capillary open, technical point to be clarified). The dissolution activity is therefore

expected to concentrate at the inner cup interface in the SGC sample. Indeed, sample cup grains may dissolve faster than loose grains due to their smaller grain size.

Could what the authors call stretching-induced fractures, be merely a localized dense network of dissolution features? Dissolution features will exhibit a different geometry in LGC sample where dissolution kinetics is expected to be smaller. According to this alternative model, reaction progress in SGC should be higher than in LGC for a given run duration, is that the case?

- **2.3/ About the Application to Nature**

In peridotites, olivine grain size is rather large (hundreds of µm). Is the LGC experiment with little stretching-induced fractures the most relevant to natural settings? In nature, the high porosity zone can be the one with the smallest grain size (e.g., cataclastic fault zone) what will happen then? I generally find that the implication for natural cases is not sufficiently discussed. Consequently, the reader has sometimes the feeling that the proposed model only applies to the described experiments with their specific design.

**3/ Specific comments**

**L48**: The notion of « olivine mineralization » is unclear. I understand "formation of olivine from a fluid" whereas I believe that the authors mean "formation of carbonates from olivine". Would not "$CO_2$ mineralization" be more appropriate here?

**Section 2.1**: There are a couple of unclear issues with respect to the experimental set-up. Did the authors use a top cap made of sintered olivine as in the 2016 paper? Does the confining pressure also apply to the sample top and bottom (no deviatoric stress?)? Is the solution isolated from the inlet capillary during experiment or is the system open in order to buffer the pore pressure? This is an important issue since it defines whether the experiments have been performed in a close or (semi)open system.

**Section 2.1**: The authors mention that they use forsterite. Is that San Carlos olivine, please clarify, since it would define the amount of ferrous iron that is present in the system.

**Section 2.1**: Can the authors exclude that drilling the aggregate to fill it with olivine sand grains may induce micro-cracks in the inner cup wall that will further localize dissolution features? Does cooling of the aggregate after sintering can induce thermal micro-cracks (nano-tomography characterization of the cup before running the sample?)?

**L125**: The present experiment only differs from the 2016 one by the change in the grain size of the sintered olivine poly-crystal that forms the cup. The importance of increasing the grain size of the cup olivine grains to approach the grain size of those located inside the cup should be more emphasized in the introduction section since it justifies writing a new paper !

**L234**: "Edge", I suppose the authors mean inner edge of the cup according to Fig. 6 in the 2016 paper. Please clarify.

**Section 3.3 and 3.4**: It is not always clear in these two sections whether the authors are describing features belonging to LGC, SGC sample or both? Please clarify.

**Section 3.3b**: Dissolution features occur to be planar and perpendicular to the vertical z-axis (Fig. 6b & 9b). Is that related to the sample geometry, stress distribution? Are they expected to develop as such in rocks? This geometry of the dissolution features is not really discussed in the manuscript although

they are the only macroscopic features that generate porosity in the sample (LGS) produced in the present study.

**L287**: "became dominate" should be "became dominant"?

**L277-288**: The grain size fluctuations described in this paragraph are fascinating. If I understand correctly, the size evolution of several grains has been monitored although only the size variation of a selected grain is displayed in Fig. 12. The possible link between local grain dissolution after 68h and development of fracturing in the cup is highly interesting. It would however be useful to see the size evolution of more than a single grain (what seems to be possible according to the available dataset?) to strengthen the inference. Would it also be possible to give error bars on the volume data to emphasize the significance of the observed variations?

**Section 4.1:** It is unclear to me what data from this study supports the notion that the tubes pierce through the whole grains (holes), could not they just remain pits.

**L331**: "Our quantitative estimates indicate that in these experiments, crystallization pressure can lead to maximum ~5% volume expansion". Is not that estimate an average value of the volume expansion over the whole sample whereas reaction crystallization-pressure induced fracturing should be regarded as a volume expansion at the local scale? Please clarify.

**L331**: I suppose that this assertion relates to the LGC, what is the volume expansion estimate for the SGC experiments where crystallization-pressure induced fracturing is also ruled out by the authors?

*Comments related to the rest of the ms are included in the general comments section.*

---

## Referee Comment (RC2) · Anonymous Referee #2 · 23 May 2018

Review of manuscript se-2018-28 entitled "Generating porosity during olivine carbonation via dissolution channels and expansion cracks", by Xing et al.

General comments:

This manuscript presents an experimental carbonation of olivine aggregates and the real-time observation of the reaction using in-situ dynamic X-ray microtomography and nanotomography. It builds on a previous study by the same group (Zhu et al., 2016) in which the authors reacted a cup made up of sintered fine-grained olivine (0-20 $\mu$m) and searched for dissolution, precipitation, and fracturing evidences. Here, the authors re use these data, perform more advanced investigations, and run a new experi-

ment using coarse-grained olivine (80-100 $\mu$m). The main consequence of the olivine grain size difference is that precipitation of magnesite is spatially heterogeneous in the fine-grained experiment while it is homogeneous in the coarse-grained experiment. In the fine-grained experiment, the heterogeneous precipitation of magnesite produces a differential volume increase between the interior (maximum increase) and the near-surface (no increase) of the cup walls so that the near-surface domains are fractured. This leads the authors to suggest that reaction-induced fracturing occurs during carbonation and helps maintaining the reactive surfaces in olivine on short time scales. The recognition of dissolution pits (etch pits) and channels indicates that dissolution could be the process maintaining the reaction on the long term.

Overall, the manuscript presented by Xing and coauthors is well written, presents interesting observations leading to logical conclusions, which make it a valuable contribution. The experimental setup uses modern technics (in-situ dynamic x-ray microtomography and nanotomography) that bring novel observations on carbonation reaction. I suspect the data processing to be heavy and to require a lot of work and efforts, which makes this study even more valuable. I therefore recommend this manuscript for publication, provided the authors clarify a few points that I detail in the following.

Specific comments:

1) This study uses previous work from Zhu et al. (2016). I assume, but am not completely sure, that the authors simply used the data already acquired and did not run a new experiment using fine-grained olivine aggregate. Is that correct? In any case, Paragraph 3.2 could be slightly modified to emphasize more clearly which part of the observations and data processing is from Zhu et al. (2016) (then in theory not the best situated in a Results section but this is not a problem here) and what is completely new.

2) Regarding the experimental setup, I guess there was a lid of olivine aggregate on the cup as in Zhu et al. (2016)? It may be worth mention it and represent it in Figure 3. Do the authors have an idea of the fluid flow direction in the cell? Is it purely vertical

or does the more porous core (loose olivine sand) involve lateral flow through the cup walls? It may be of interest to explain and understand magnesite precipitation or non-precipitation in the different domains.

3) The authors report a 10% initial porosity of the cup wall (line 129). Is it identical in the two cups? I wonder what makes this porosity, is it olivine grain boundaries or fractures? How interconnected is this porosity? I would suggest adding a short paragraph describing the structure of the starting aggregate.

4) This is of importance because I do not completely get the distinction between the fracture and the dissolution planes, particularly the lines 239-243 and the Figures 5 and 6. For the reaction-induced fractures, do they cut across the olivine grains (i.e., breaking them in two) or do they use the grain boundaries? I have the same question for the dissolution planes. I also wonder why dissolution would form a single flat plane and not an anastomosing network. What causes the dissolution plane to have this geometry? Can we imagine that the size of the olivine grains plays a role, favoring large-scale, single structures (e.g., what is described as a single dissolution plane) in the coarse-grained experiment and small-scale, network-like structures (e.g., what is described as a crack pattern) in the fine-grained experiment?

5) The last part of the discussion focuses on the application of the findings to natural systems. I think this could be improved by discussing more how the results compare with observations made on natural samples. For instance, in the last paragraph, the authors state that reaction-induced fracturing helps maintaining the reaction on the short term while dissolution does it on the long term. There is one study on natural samples that could reinforce these conclusions. Reaction-induced fracturing has been recognized in peridotites serpentinized at mid-ocean ridges by Rouméjon and Cannat (2014, G3). The hydration leading to the replacement of the olivine by serpentine occurs along a network of fractures (forming the so-called mesh texture). These fractures develop in two steps: 1) conjugate fracture planes of combined tectonic and thermal contraction origin that crosscut the olivine before hydration; 2) reaction-induced fractures associated to the volume increase consecutive to serpentinization while hydration occurs. It is shown (see their Figure 8c) that the reaction-induced fracturing occurs in the early stages of serpentinization (probably before 20% of serpentinization) while the rest of the volume increase is accommodated by the serpentine itself and dissolution processes dominate until completion of the reaction.

6) Another question to develop concerns the typical length scale of the dissolution and fracturing processes. In this study, such processes occur at nano- to micrometer scales (Lines 291-2923: "micro-meter scale in the case of the coarse-grained aggregate and at nano-meter scale in the case of fine-grained aggregate, and reaction-induced fracturing in the case of the fine-grained aggregate."). These scales are rather small for natural samples and would correspond to a second order permeability. Much larger permeability pathways (e.g., mm to cm cracks) are required to efficiently channel fluids and provoke carbonation of significant volumes at rapid time scales. Do the authors think their results are transposable at such larger scales? And if so, could they make suggestions on what would it require for actual CO2 sequestration? (e.g., system dimensions, typical grain size, . . .)

7) Finally, I find Figure 12 intriguing and full of potential for further studies. This is a nice example of advances made possible by X-ray tomography in the comprehension of mineralogical reactions. I think there is still a lot of microstructural work possible using such technics.

Technical corrections:

Lines 19-20: "dissolution fractures developed". I guess the authors mean dissolution planes?

Line 24: I would rephrase the end of this sentence: "by the volume mismatch in the cup walls, between the expanding interior and the near-surface that keeps a nearly constant volume"?

Line 32: Not sure Escartin et al. (1997) is the most relevant here. Maybe you could cite review papers such as Deschamps et al. (2013, Lithos) or Guillot et al. (2015, Tectonophysics) that list and discuss peridotite exposures

Line 78: "olivine mineralization" is unclear, needs clarification

Lines 104-105: I suggest reformulation: "with larger grain size (80-100 $\mu$m) compared to the previous experiment reported by Zhu et al. (2016; 0-20 $\mu$m).

Line 129: I guess the 10% porosity refers to the coarse-grained olivine aggregate. Is it comparable to the fine-grained olivine aggregate?

Line 133: I do not think the authors clearly mention the duration of their experiment. I suppose this is 36h for the coarse-grained experiment and 7 days for the fine-grained experiment (from lines 149 and 150)? In the abstract they mention "until the olivine aggregates became disintegrated". Is it really the case? It should be described here.

Line 164: "simplified analyses" is a bit odd. I think I understand what the authors mean but that can be rephrased.

Lines 168-169: "both the cup wall (surface?) and the cup (wall?) interior"

Lines 187-189: Not sure this paragraph is really useful

Line 191: "is not observed to be dominated by stress-generated fracturing" is a though formulation, needs rephrasing

Lines 192-193: this should go in the discussion

Lines 193-195: Needs rephrasing, the sentence about the loose grains seems to be in the middle of two sentences talking about the cup walls. "precipitation-caused non-uniform stretching" is hard to follow.

Line 195: "in the sample". I guess the authors refer to inside the olivine aggregate as opposed to the surface. Throughout the manuscript, it is sometimes hard to follow what

the authors are referring to due to the changes in the terminology (e.g., interior is also used to refer to the inside of the cup wall)

Line 199: "as a single plane" instead of "along a main plane"?

Line 202: "disintegration of the cup's wall"

Lines 203-204: Maybe I am wrong, but "fragile" and "cohesion-less" seem to say the same thing here, so I suggest rephrasing

Line 205: "after 68 hours of reaction". "exhibits a hierarchical manner" is a bit odd, needs rephrasing.

Lines 206-207: "Figure 5b shows that the fractures first occurred in areas close to the surface and propagated inwards. "The fracture first developed as a single...".

Line 209: "systematic" can be deleted

Lines 210-212: The end of the sentence is not clear and would need rephrasing. But it also looks like discussion and should be removed (as well as line 213).

Line 215: "their", not really clear what it refers to

Line 216: cite Figure 8 here instead of at the end of the next sentence?

Line 231: "a volume expansion"

Line 241: "It's obvious"...not really

Line 242: "along"

Line 251: "exhibits a hierarchical geometry in which the fractures that appeared first are now the largest"

Line 252: "domains" instead of "patches"

Line 262: Is there a way to have typical sizes (e.g., diameters) or it is too variable?

Line 326: I would add a reference here, as in the introduction (line 51)

Line 360: "should still be in a range"

Line 422: "and the resulting contrast in the expansion" is unclear, needs rephrasing

Figure 1: "reacted" and "unreacted" are not clearly visible, change the color Lines 579-580: the sentence is complicated, needs reformulation

Figure 4: It took me a while before understanding where the subvolume 2 was exactly positioned. It gives the impression that the subvolume 2 was outside of the cup. Perhaps a 2D sketch would be more efficient.

Figure 9: If possible, add the orientation of these volumes

Figure 10: To which experiment and time does this figure correspond? Also, I could understand that the dashed line polygon corresponds to the upper half of the photos but it was not straightforward. I suggest indicating that differently (e.g., annotate the upper half of the photos or modify the orientation picture).

---

## Referee Comment (RC3) · Anonymous Referee #3 · 25 May 2018

The manuscript presents experimental results for carbonation of olivine in 4D, three spatial dimensions plus time. It is a follow-up on a previous paper from the same group (Zhu et al., 2016), and provides both some additional data on the experiment in the previous paper and results from a new experiment using a coarser-grained initial material. There is not much 4D data on such processes available in the literature, which makes this a topic that is suitable for publication, within the scope of the journal, and will attract a lot of attention. The paper is also well written and fairly easy to read, although there are some misprints.

However, in my opinion the authors are spending too much time on repeating state-

ments and data that is already present in Zhu et al. (2016). Of course some background information from the previous paper needs to be included, but quite large parts of the text can be removed and replaced with a reference to Zhu et al., and perhaps more importantly, repeated background information should be clearly marked as being repeated, to avoid giving the fake impression of being new data presented in this manuscript. Furthermore, there is not that much information about the new coarse-grained experiment, and it would be a lot more interesting to see some more details about the differences between the fine-grained and coarse-grained experiments instead of an extended discussion of the crack patterns presented by Zhu et al. Thus, I recommend a major revision where the authors should reduce the amount of repeated data. In the following I firstly give my recommendations for what should be removed, and suggest some other data that could be included. Then I list a few major concerns, followed by some minor comments and a list of misprints.

Remove or add

- Section 3.2 is mainly repeated from Zhu et al., and should be shortened significantly. There is no need to repeat the entire description of the formed cracks and cemented patches, since the interested reader can look up Zhu et al. instead. Figure 7 is a direct repetition from Zhu et al., but I admit that this might be useful to include as background. The grey value distribution in figure 8 is also included in Zhu et al., although in the supplementary material and without the best fit representation. If the authors include the same type of analysis on the coarse-grained experiment, this would be interesting, but without it I don't see much value in the figure. In the end of the section the authors estimate the expansion, something Zhu et al. didn't do, apart from noting that the material expanded. It is fine to include this number here, but it would be more interesting with a similar number from the coarse-grained experiment. The expansion might be zero in that case, but if so it should be stated clearly. Furthermore, is it possible to extract expansion as function of time from the data? That might be interesting.

- Section 3.4 would be more useful if it compared data from the fine-grained experiment

with similar data from the coarse-grained experiment. As it is, this section is mainly a more verbose version of what is already written by Zhu et al., with a few additional estimates of growth rate.

- Section 4.2 is a large chunk of the discussion, and mainly repeats stuff from Zhu et al., and most of it can be removed. Also, figure 13 can safely be removed. Although it shows 3D data and Zhu et al. only presented a 2D plot, the data here is hard to interpret and only discussed in context of 2D porosity distribution. Thus, it has little value. A few comparisons between the two experiments might be interesting, but not much more.

Major concerns

- In figure 6, it is stated that the linear feature in the coarse-grained aggregate is caused by dissolution. It is not at all clear to me why dissolution would cause such a linear structure. In a flow-through experiment you might expect some sort of wormholing, but this can hardly be relevant here. Rather, I would assume that what is shown is a single axial crack, with secondary dissolution of the crack faces. If the authors really think the structure is just caused by dissolution, they have some explaining to do as to why this ends up being linear. Some data on formation of this crack-like structure in time and perhaps tracking of grains at each surface of the structure might help. Now this is just guessing, but I think such a structure might be caused by a crack if you have less, but non-zero, volume expansion in the coarse-grained aggregate. Higher volume change would naturally then lead to a denser crack pattern. Of course there is some dissolution going on, but I have a hard time understanding why it would organize itself as a crack.

- I do not believe that the "expansion cracks via stretching" mechanism is a reasonable full explanation of the cracks. On lines 231–233 the authors state that grains in the center of the cup wall move apart. Now, these grains were initially bonded mechanically. How can they separate if these bonds are not broken during the process?

Clearly they must be. This might be caused by some sort of dissolution-precipitation creep or by reaction induced cracking. I guess it would be difficult to tell the difference based on the available data, but to me it seems likely that these bonds are at least partly cracked, before the crack is recemented by the reaction products. Thus, there might be dense, invisible cracking in the center of the cup wall, while the effect of this cracking and expansion in the wall center is a less dense crack pattern on the outside of the cup wall. A rock simply cannot expand in a chemical process that involves dissolution of the base material and precipitation of some product unless bonds between the initial grains are broken. Uneven heating of a rock would cause something like the situation presented in figure 14, with the yellow part being warmer than the green, but in a chemical process there has to be some deformation in the reacted part.

Minor comments

Generally, please state more clearly in figure captions whether results are from the fine-grained or coarse-grained experiment.

Line 179: A description of the color scale used would be helpful, e.g. something like "... where X represents black, and Y is white".

Lines 264–266: Here, it is noted that there is evidence for hierarchical fracturing within the olivine grains. Later, it is peculiarly written on lines 333–335 that there is no evidence of cracks in olivine grains. This is at best sloppy. Furthermore, why are these cracks forming, if not by the very reaction induced cracking that the authors claim is not observed?

Lines 278–288, and figure 12: It would be interesting to see plotted the individual volume change of the olivine grain and the formed precipitates. I would also suggest changing the figure a bit, it is hard to see the structure of the precipitated material. Something closer to figure 5 in Zhu et al. would be better.

Figure 1: The text "Reacted" is extremely hard to see, please consider changing the

color.

Figure 5: Why is the dissolution or crack always in the lower right corner?

Figure 9a: I'm unable to interpret the fracture network, please consider reworking the figure a bit and perhaps include a view from different angles.

Misprints

Line 38: reaction -> reactions

Line 92: system -> systems

Line 131 (and other places): x-ray -> X-ray

Line 154: images -> image

Line 206: shown -> shows

Line 232: gains -> grains

Line 242: alone -> along

Line 251: appeared -> appearing

Line 255 and figure 14 caption: none-uniform -> non-uniform

Figure 1: crystalization -> crystallization

References:

Zhu, Wenlu; Fusseis, Florian; Lisabeth, Harrison; Xing, Tiange; Xiao, Xianghui; De Andrade, Vincent; Karato, Shun-ichiro (2016), Experimental evidence of reaction-induced fracturing during olivine carbonation, Geophys. Res. Lett. 43 (18), pages 9535–9543

---

## Editor Comment (EC1) · M. J. Heap (Editor) · 31 May 2018

Dear authors,

As you can see, I've now received three reviews of your manuscript: "Generating porosity during olivine carbonation via dissolution channels and expansion cracks". The reviews are generally positive. If you're willing, please now prepare a point-by-point rebuttal letter to address the concerns of the three reviewers and a revised manuscript (preferably with the changes highlighted). In your revised manuscript, please pay particular attention to explicitly highlight the similarities/differences between this manuscript and Zhu et al. (2016).

[Figure]

Thanks for submitting your work to Solid Earth.

Mike Heap (Topical Editor of Solid Earth)

---

## Author Comment (AC1) · 15 Jun 2018

June 14, 2018

Dear Editor,

We thank the reviewers for their comments. In the rebuttal letter, we addressed each reviewer's comments separately. The reviewers' comments are italicized, followed by our point-by-point response to the reviewers.

Enclosed please also find a copy of the revised manuscript with changes highlighted.

Sincerely,

[Figure]

Tiange Xing

Please also note the supplement to this comment:
https://www.solid-earth-discuss.net/se-2018-28/se-2018-28-AC1-supplement.zip

—————————————————

---

## Author Comment (AC2) · 15 Jun 2018

June 14, 2018

Dear Reviewer,

We thank you for your comments. In the rebuttal letter, we addressed your comments separately. The comments are italicized, followed by our point-by-point response.

Enclosed please also find a copy of the revised manuscript with changes highlighted.

Sincerely,

[Figure]

Tiange Xing

Please also note the supplement to this comment:
https://www.solid-earth-discuss.net/se-2018-28/se-2018-28-AC2-supplement.zip

---

## Author Response (AR1)

June 14, 2018

Dear Editor,

We thank the reviewers for their comments. In the rebuttal letter, we addressed each reviewer's comments separately. The reviewers' comments are italicized, followed by our point-by-point response to the reviewers.

Enclosed please also find a copy of the revised manuscript with changes highlighted.

Sincerely,

**Tiange Xing**

**Anonymous Referee #1**

1/Introduction

The manuscript by Xing et al. entitled "Generating porosity during olivine carbonation via dissolution channels and expansion cracks" reports on a very nice piece of experimental work on the in-situ hydrothermal carbonation of olivine aggregates. Careful attention is paid to the real-time development of microstructures to unravel reaction-induced porosity changes and fracturing. Indeed coupling and feedbacks between dissolution/crystallization and generation of new fluid pathways within mineral aggregates (synthetic rock) is still poorly known although highly relevant to metamorphic and alteration reactions which involve aqueous and carbonate fluids. The study basically confirms a reaction-transport-deformation model that has been proposed by the same authors in 2016 based on a very similar experiment using the same characterization technique. The difference with the present study mostly relies on the use of a different mineral grain size. The high similarity between the two studies makes sometimes difficult to distinguish between data that have been collected here and in the previous study. Naming samples like LGC (larger grain cup) and SGC (smaller grain cup) would potentially help.

Thank you for the suggestion! We adopted the name SGC for the fine-grained cup and LGC for coarsegrained cup in the revised manuscript.

2/ General comments

**- 2.1 About Reaction progress**

Estimate of the overall reaction progress in the cup is an important piece of information. In a system that is prone to porosity clogging due to volume expansion of the solid phases, it is expected that the nature, density and geometry of fluid pathways will change with reaction progress. Basically, are the features described in this study relevant to peridotites that are at the beginning of the carbonation process (

Section 4.1: It is unclear to me what data from this study supports the notion that the tubes pierce through the whole grains (holes), could not they just remain pits.

From the 3D analysis of the nanotomographic data, some tubes have clear bottoms inside the grains, but others form through-going holes across the whole grains. Since we do not see any precipitation filling or clogging the tubes, it is reasonable to think that these features represent different stages of tube formation. They started as the etch-pits. At the pits site, enhanced dissolution takes place, and the pits grow deeper and eventually become the through-going tubes.

We clarified this point in the text (lines 290-293)

L331: "Our quantitative estimates indicate that in these experiments, crystallization pressure can lead to maximum ~5% volume expansion". Is not that estimate an average value of the volume expansion over the whole sample whereas reaction crystallization-pressure induced fracturing should be regarded as a volume expansion at the local scale? Please clarify.

Here the "crystallization pressure" means the mechanical force produced by the chemical reaction. We modify the sentence to clarify this point: "Our quantitative estimates indicate that in these experiments, the maximum volume expansion is ~5%."

L331: I suppose that this assertion relates to the LGC, what is the volume expansion estimate for the SGC experiments where crystallization-pressure induced fracturing is also ruled out by the authors?

This above volume expansion estimate is for the SGC experiment. For the LGC experiment, no detectable volume expansion is observed.

**Anonymous Referee #2**

**General comments:**

This manuscript presents an experimental carbonation of olivine aggregates and the real-time observation of the reaction using in-situ dynamic X-ray microtomography and nanotomography. It builds on a previous study by the same group (Zhu et al., 2016) in which the authors reacted a cup made up of sintered finegrained olivine (0-20  $\mu$ m) and searched for dissolution, precipitation, and fracturing evidences. Here, the authors re use these data, perform more advanced investigations, and run a new experiment using coarsegrained olivine (80-100  $\mu$ m). The main consequence of the olivine grain size difference is that precipitation of magnesite is spatially heterogeneous in the fine-grained experiment while it is homogeneous in the coarse-grained experiment. In the fine-grained experiment, the heterogeneous precipitation of magnesite produces a differential volume increase between the interior (maximum increase) and the nearsurface (no increase) of the cup walls so that the near-surface domains are fractured.

This leads the authors to suggest that reaction-induced fracturing occurs during carbonation and helps maintaining the reactive surfaces in olivine on short time scales.

*The recognition of dissolution pits (etch pits) and channels indicates that dissolution could be the process maintaining the reaction on the long term.*

*Overall, the manuscript presented by Xing and coauthors is well written, presents interesting observations leading to logical conclusions, which make it a valuable contribution.*

The experimental setup uses modern technics (in-situ dynamic x-ray microtomography and nanotomography) that bring novel observations on carbonation reaction. I suspect the data processing to be heavy and to require a lot of work and efforts, which makes this study even more valuable. I therefore recommend this manuscript for publication, provided the authors clarify a few points that I detail in the following.

**Specific comments:**

1) This study uses previous work from Zhu et al. (2016). I assume, but am not completely sure, that the authors simply used the data already acquired and did not run a new experiment using fine-grained olivine aggregate. Is that correct? In any case, Paragraph 3.2 could be slightly modified to emphasize more clearly which part of the observations and data processing is from Zhu et al. (2016) (then in theory not the best situated in a Results section but this is not a problem here) and what is completely new.

The microtomography experiment on the fine-grained olivine cup is the same as in Zhu et al. (2016) but the 3D analyses of fracture network, quantification of grain volume and porosity are new. The nanotomography experimental results (dissolution pits and tubes) on the reacted fine-grained cup are completely new.

Zhu et al. (2006) hypothesized that large grains would be preferred sites for precipitation of new crystals, Thus the loose olivine grains (100-500  $\mu$ m) in the immediate vicinity of the inner cup surface (made of fined-grained, 1-20  $\mu$ m, olivine) function as precipitate traps and thereby kept precipitation level at the surface of the olivine cup wall low. This led to the contrast in magnesite precipitation within the cup wall.

Because the grain size contrast played a key role in generating non-uniform volume expansion, we do need to compare the results of the fine-grained experiment (Zhu et al., 2016) to the new coarse-grained

experiments. To make the points clearer, we now use abbreviations SGC (for the fine-grained experiment) and LGC (for the coarse-grained experiment) at another reviewer's suggestion.

We have edited the text and made the new results more explicit.

2) Regarding the experimental setup, I guess there was a lid of olivine aggregate on the cup as in Zhu et al. (2016)? It may be worth mention it and represent it in Figure 3. Do the authors have an idea of the fluid flow direction in the cell? Is it purely vertical or does the more porous core (loose olivine sand) involve lateral flow through the cup walls? It may be of interest to explain and understand magnesite precipitation or nonprecipitation in the different domains.

We modified Figure 3 to add the lid.

These were not flow-through experiments. A constant pore pressure of 10 MPa was kept during the experiment. The pore fluid does not flow because there was no pore pressure gradient along or across the sample. The whole sample assembly was fully-saturated.

We added this information in Section 2.1 (lines 141-143).

**3) The authors report a 10% initial porosity of the cup wall (line 129). Is it identical in the two cups? I wonder what makes this porosity, is it olivine grain boundaries or fractures? How interconnected is this porosity? I would suggest adding a short paragraph describing the structure of the starting aggregate.**

The initial porosities of the two aggregated are similar, estimated from the microtomographic scans (see new Figure 8). From these scans, the pores the coarse-grained aggregate are fully connected, whereas the pores in the fine-grained aggregates form different connected clusters. This might also be partly because of that some of the pore throat are beyond the resolution of the microtomographic data. Both aggregates are very permeable, indicating well-connected pore networks.

We added more descriptions of the starting materials in the text (lines 128-138).

4) This is of importance because I do not completely get the distinction between the fracture and the dissolution planes, particularly the lines 239-243 and the Figures 5 and 6. For the reaction-induced fractures, do they cut across the olivine grains (i.e., breaking them in two) or do they use the grain boundaries? I have the same question for the dissolution planes. I also wonder why dissolution would form a single flat plane and not an anastomosing network. What causes the dissolution plane to have this geometry? Can we imagine that the size of the olivine grains plays a role, favoring large-scale, single structures (e.g., what is described as a single dissolution plane) in the coarse-grained experiment and small-scale, network-like structures (e.g., what is described as a crack pattern) in the fine-grained experiment?

The reaction-induced fractures cut through cluster of grains. Because the spatial resolution in the microtomography experiments is ~2 micron, and the reaction-induced fractures occur only in the fine-grained sample (0-20 micron), we are not able to resolve whether the grains were broken in two , most fractures developed using grain boundaries. Nanotomography data show that some fractures also cut though the olivine grains.

The Figure 9 (now Figure 10 in the revised manuscript) shows that the dissolution plane is primarily a single feature with a few small branches. Development of such secondary features are probably limited by the reaction duration (30 hours).

We think these fractures are dissolution-assisted fractures under tri-axial extension. We explained the formation of the planar features in the text:

"Under a constant confining pressure, volume reduction in olivine grains (i.e., dissolution) likely shortened the LGC sample length as reaction proceeded. Because the axial piston was kept at a fixed position during the experiment, this shortening in sample length resulted a decrease in axial stress. Because the LGC sample is mechanically weak (less cohesion), even though the reduction in axial stress is small, it could be sufficient to cause fracture LGC in the manner of dilation bands under triaxial extension (e.g., Zhu et al., 1997). Detailed examination of the 3D images revealed the disappearance of small grains along the plane which is clear evidence of dissolution. Thus we refer to these planar cracks as dissolution-assisted fractures under triaxial extension. The dissolution-assisted fractures were not observed in the SGC sample because it is much stronger owing to its fine grain size (e.g. Eberhardt et al., 1999; Singh, 1988). The triaxial extension stress condition would be no longer present once precipitation started (after ~36 hours) and sample volume expansion took place."

5) The last part of the discussion focuses on the application of the findings to natural systems. I think this could be improved by discussing more how the results compare with observations made on natural samples. For instance, in the last paragraph, the authors state that reaction-induced fracturing helps maintaining the reaction on the short term while dissolution does it on the long term. There is one study on natural samples that could reinforce these conclusions. Reaction-induced fracturing has been recognized in peridotites serpentinized at mid-ocean ridges by Rouméjon and Cannat (2014, G3). The hydration leading to the replacement of the olivine by serpentine occurs along a network of fractures (forming the so-called mesh texture). These fractures develop in two steps: 1) conjugate fracture planes of combined tectonic and thermal contraction origin that crosscut the olivine before hydration; 2) reaction-induced fractures associated to the volume increase consecutive to serpentinization while hydration occurs. It is shown (see their Figure 8c) that the reaction-induced fracturing occurs in the early stages of serpentinization (probably before 20% of serpentinization) while the rest of the volume increase is accommodated by the serpentine itself and dissolution processes dominate until completion of the reaction.

Many thanks for the suggestion and we also really appreciate the reference. This could be an important direction for the future work.

We added new discussion on the relevance of the laboratory experiments and field studies (lines 450-458).

6) Another question to develop concerns the typical length scale of the dissolution and fracturing processes. In this study, such processes occur at nano- to micrometer scales (Lines 291-2923: "micro-meter scale in the case of the coarse-grained aggregate and at nano-meter scale in the case of fine-grained aggregate, and reaction-induced fracturing in the case of the fine-grained aggregate."). These scales are rather small for natural samples and would correspond to a second order permeability. Much larger permeability pathways (e.g., mm to cm cracks) are required to efficiently channel fluids and provoke carbonation of significant volumes at rapid time scales. Do the authors think their results are transposable at such larger

**scales? And if so, could they make suggestions on what would it require for actual CO2 sequestration? (e.g., system dimensions, typical grain size, . . .)**

In this study, we focus on understanding the underlying mechanism of porosity generation during olivine carbonation reaction. The physics of the porosity generation mechanism is scale independent. While the grain size used in laboratory settings are much smaller, the time scale using is also much shorter. It is conceivable that given enough time, the nano- to micrometer scale cracks could grow to centimeter fractures.

We do however, recognize that upscaling is always challenging for laboratory investigation. We added new discussion on the relevance of the laboratory experiments and field studies (lines 450-458).

7) Finally, I find Figure 12 intriguing and full of potential for further studies. This is a nice example of advances made posible by X-ray tomography in the comprehension of mineralogical reactions. I think there is still a lot of microstructural work possible using such technics.

We agree.

Technical corrections:

Lines 19-20: "dissolution fractures developed". I guess the authors mean dissolution planes?

Corrected.

Line 24: I would rephrase the end of this sentence: "by the volume mismatch in the cup walls, between the expanding interior and the near-surface that keeps a nearly constant volume"?

We have modified the sentence to clarify.

Line 32: Not sure Escartin et al. (1997) is the most relevant here. Maybe you could cite review papers such as Deschamps et al. (2013, Lithos) or Guillot et al. (2015, Tectonophysics) that list and discuss peridotite exposures

We have added the citation of Deschamps et al. (2013) in the introduction.

Line 78: "olivine mineralization" is unclear, needs clarification

We have modified this to olivine carbonation.

Lines 104-105: I suggest reformulation: "with larger grain size (80-100  $\mu$ m) compared to the previous experiment reported by Zhu et al. (2016; 0-20  $\mu$ m).

Modified.

*Line 129: I guess the 10% porosity refers to the coarse-grained olivine aggregate. Is it comparable to the fine-grained olivine aggregate?*

Both samples have an initial porosity of 10%. Also shown in the Figure 8, initial porosity of the two samples are comparable.

*Line 133: I do not think the authors clearly mention the duration of their experiment. I suppose this is 36h for the coarse-grained experiment and 7 days for the fine-grained experiment (from lines 149 and 150)?*

In the abstract they mention "until the olivine aggregates became disintegrated". Is it really the case? It should be described here.

The clarification has been added.

*Line 164: "simplified analyses" is a bit odd. I think I understand what the authors mean but that can be rephrased.*

This phrasing has been modified.

Lines 168-169: "both the cup wall (surface?) and the cup (wall?) interior"

The sentence has been modified to clarify this.

Lines 187-189: Not sure this paragraph is really useful

We have modified the paragraph and also the structure of the section 3.

*Line 191: "is not observed to be dominated by stress-generated fracturing" is a though formulation, needs rephrasing*

Modified.

Lines 192-193: this should go in the discussion

We have incorporate that into the discussion.

*Lines 193-195: Needs rephrasing, the sentence about the loose grains seems to be in the middle of two sentences talking about the cup walls. "precipitation-caused nonuniform stretching" is hard to follow.*

Modified.

Line 195: "in the sample". I guess the authors refer to inside the olivine aggregate as opposed to the surface. Throughout the manuscript, it is sometimes hard to follow what the authors are referring to due to the changes in the terminology (e.g., interior is also used to refer to the inside of the cup wall)

Modified.

Line 199: "as a single plane" instead of "along a main plane"?

Corrected.

Line 202: "disintegration of the cup's wall"

**Modified.**

Lines 203-204: Maybe I am wrong, but "fragile" and "cohesion-less" seem to say the same thing here, so I suggest rephrasing

Modified.

Line 205: "after 68 hours of reaction". "exhibits a hierarchical manner" is a bit odd, needs rephrasing.

Modified.

*Lines 206-207: "Figure 5b shows that the fractures first occurred in areas close to the surface and propagated inwards. "The fracture first developed as a single..."*.

Modified.

*Line 209: "systematic" can be deleted*

Modified.

*Lines 210-212: The end of the sentence is not clear and would need rephrasing. But it also looks like discussion and should be removed (as well as line 213).*

This is a description of the fractures network pattern. Its morphology is similar to the 'mud desccication crack'.

Line 215: "their", not really clear what it refers to

We have modified the sentence to clarify.

Line 216: cite Figure 8 here instead of at the end of the next sentence?

Added the citation.

Line 231: "a volume expansion"

Modified.

*Line 241: "It's obvious"...not really*

A citation to Figure has been added here.

Line 242: "along"

Corrected.

Line 251: "exhibits a hierarchical geometry in which the fractures that appeared first are now the largest"

Line 252: "domains" instead of "patches"

Modified.

Line 262: Is there a way to have typical sizes (e.g., diameters) or it is too variable?

The typical radius of the ~6 pixels (360 nm). We discussed the radius in the estimation of the permeability in discussion. Statement on the inner diameter of the tubes has been added here.

Line 326: I would add a reference here, as in the introduction (line 51)

Line 360: "should still be in a range"

Corrected.

Line 422: "and the resulting contrast in the expansion" is unclear, needs rephrasing

The statement has be modified to clarify.

*Figure 1: "reacted" and "unreacted" are not clearly visible, change the color Lines 579-580: the sentence is complicated, needs reformulation*

**Modified.**

*Figure 4: It took me a while before understanding where the subvolume 2 was exactly positioned. It gives the impression that the subvolume 2 was outside of the cup. Perhaps a 2D sketch would be more efficient.*

We changed the shading to better illustrate the cubes. The 2D representation of subvolumes 1&2 could be found in Zhu et al. (2016).

Figure 9: If possible, add the orientation of these volumes

We have added annotation to indicate the cup's outer and inner surface (now Figure 10).

Figure 10: To which experiment and time does this figure correspond? Also, I could understand that the dashed line polygon corresponds to the upper half of the photos but it was not straightforward. I suggest indicating that differently (e.g., annotate the upper half of the photos or modify the orientation picture).

We have added in the figure captions the names SGC for the fine-grained sample and LGC for the coarsegrained sample.

We modified figure capture to better describe the dashed lines (new Figure 11).

**Anonymous Referee #3**

The manuscript presents experimental results for carbonation of olivine in 4D, three spatial dimensions plus time. It is a follow-up on a previous paper from the same group (Zhu et al., 2016), and provides both some additional data on the experiment in the previous paper and results from a new experiment using a coarser-grained initial material. There is not much 4D data on such processes available in the literature, which makes this a topic that is suitable for publication, within the scope of the journal, and will attract a lot of attention. The paper is also well written and fairly easy to read, although there are some misprints.

However, in my opinion the authors are spending too much time on repeating statements and data that is already present in Zhu et al. (2016). Of course some background information from the previous paper needs to be included, but quite large parts of the text can be removed and replaced with a reference to Zhu et al., and perhaps more importantly, repeated background information should be clearly marked as being repeated, to avoid giving the fake impression of being new data presented in this manuscript. Furthermore, there is not that much information about the new coarse-grained experiment, and it would be a lot more interesting to see some more details about the differences between the fine-grained and coarse-grained experiments instead of an extended discussion of the crack patterns presented by Zhu et al. Thus, I recommend a major revision where the authors should be removed, and suggest some other data that could be included. Then I list a few major concerns, followed by some minor comments and a list of misprints.

We made a major revision in Section 3. Detailed changes are listed below.

Remove or add

- Section 3.2 is mainly repeated from Zhu et al., and should be shortened significantly.

There is no need to repeat the entire description of the formed cracks and cemented patches, since the interested reader can look up Zhu et al. instead. Figure 7 is a direct repetition from Zhu et al., but I admit that this might be useful to include as background.

The grey value distribution in figure 8 is also included in Zhu et al., although in the supplementary material and without the best fit representation. If the authors include the same type of analysis on the coarsegrained experiment, this would be interesting, but without it I don't see much value in the figure. In the end of the section the authors estimate the expansion, something Zhu et al. didn't do, apart from noting that the material expanded. It is fine to include this number here, but it would be more interesting with a similar number from the coarse-grained experiment. The expansion might be zero in that case, but if so it should be stated clearly. Furthermore, is it possible to extract expansion as function of time from the data? That might be interesting.

We have reorganized the section 3 and to make our new result more explicit. Paragraphs and figure that might be considered a repetition have been removed from the section.

For the grey value histogram, we have now included our analysis on the LGC sample in the new Figure 7.

The expansion is not observed in the coarse-grained experiment. We have clarified this in the revised paper (lines 226-227).

- Section 3.4 would be more useful if it compared data from the fine-grained experiment with similar data from the coarse-grained experiment. As it is, this section is mainly a more verbose version of what is already written by Zhu et al., with a few additional estimates of growth rate.

This section is modified in the revised manuscript (lines 265-287)

- Section 4.2 is a large chunk of the discussion, and mainly repeats stuff from Zhu et al., and most of it can be removed. Also, figure 13 can safely be removed. Although it shows 3D data and Zhu et al. only presented a 2D plot, the data here is hard to interpret and only discussed in context of 2D porosity distribution. Thus, it has little value. A few comparisons between the two experiments might be interesting, but not much more.

We do think that it is important to visualize the 3D porosity distribution (now Figure 14). The figure provide an easy visualization that 1) the porosity reduction as a result of precipitation is non-uniform (with smaller porosity at the center); 2) there is no detectable secondary porosity generated in the region where most precipitation takes place. This is the key difference between the "crystallization pressure model" and the "volume mismatch model".

We modified the section 4.2 in the revised manuscript.

**Major concerns**

- In figure 6, it is stated that the linear feature in the coarse-grained aggregate is caused by dissolution. It is not at all clear to me why dissolution would cause such a linear structure. In a flow-through experiment you might expect some sort of wormholing, but this can hardly be relevant here. Rather, I would assume that what is shown is a single axial crack, with secondary dissolution of the crack faces. If the authors really think the structure is just caused by dissolution, they have some explaining to do as to why this ends up being linear. Some data on formation of this crack-like structure in time and perhaps tracking of grains at each surface of the structure might help. Now this is just guessing, but I think such a structure might be caused by a crack if you have less, but non-zero, volume expansion in the coarse-grained aggregate. Higher volume change would naturally then lead to a denser crack pattern. Of course there is some dissolution going on, but I have a hard time understanding why it would organize itself as a crack.

Thanks for pointing this point.

We agree that these fractures are axial cracks, with secondary dissolution of the crack faces. They are renamed as "dissolution-assisted fractures under tri-axial extension". We explained the formation of the planar features in the text:

"Under a constant confining pressure, volume reduction in olivine grains (i.e., dissolution) likely shortened the LGC sample length as reaction proceeded. Because the axial piston was kept at a fixed position during the experiment, this shortening in sample length resulted a decrease in axial stress. Because the LGC sample is mechanically weak (less cohesion), even though the reduction in axial stress is small, it could be sufficient to cause fracture LGC in the manner of dilation bands under triaxial extension (e.g., Zhu et al., 1997). Detailed examination of the 3D images revealed the disappearance of small grains along the plane which is clear evidence of dissolution. Thus we refer to these planar cracks as dissolution-assisted fractures under triaxial extension. The dissolution-assisted fractures were not observed in the SGC sample because it is much stronger owing to its fine grain size (e.g. Eberhardt et al., 1999; Singh, 1988). The triaxial extension stress condition would be no longer present once precipitation started (after ~36 hours) and sample volume expansion took place."

- I do not believe that the "expansion cracks via stretching" mechanism is a reasonable full explanation of the cracks. On lines 231–233 the authors state that grains in the center of the cup wall move apart. Now, these grains were initially bonded mechanically. How can they separate if these bonds are not broken during the process?

Clearly they must be. This might be caused by some sort of dissolution-precipitation creep or by reaction induced cracking. I guess it would be difficult to tell the difference based on the available data, but to me it seems likely that these bonds are at least partly cracked, before the crack is recemented by the reaction products. Thus, there might be dense, invisible cracking in the center of the cup wall, while the effect of this cracking and expansion in the wall center is a less dense crack pattern on the outside of the cup wall. A rock simply cannot expand in a chemical process that involves dissolution of the base material and precipitation of some product unless bonds between the initial grains are broken. Uneven heating of a rock would cause something like the situation presented in figure 14, with the yellow part being warmer than the green, but in a chemical process there has to be some deformation in the reacted part.

The microtomographic images clearly show a volume expansion in the center of the cup wall where there is no evidence of cracking or porosity increase. So we think that the expansion is more likely resulted from a dissolution assisted creep process, not by fracturing caused by crystallization pressure.

The main point of Figure 15 is that in a system where the crystallization force is not large enough to directly fracture the host rock, if the volume expansion (by creep) is heterogeneous within the sample, reaction-induced fracturing and porosity increase can still occur as a result of stretching caused by the volume mismatch.

We modified the text to clarify this point (lines 444-450).

**Minor comments**

Generally, please state more clearly in figure captions whether results are from the fine-grained or coarsegrained experiment.

We have added in the figure captions the names SGC for the fine-grained sample and LGC for the coarsegrained sample.

*Line 179: A description of the color scale used would be helpful, e.g. something like "… where X represents black, and Y is white".*

This is a statement of the binarization process. We convert the image into the phase of interest (assigned value 1) and the matrix (assigned value 0).

In our data, the black to dark grey colors represent pores, and white to light grey colors represent olivine (illustrated in Figure 7).

Lines 264–266: Here, it is noted that there is evidence for hierarchical fracturing within the olivine grains. Later, it is peculiarly written on lines 333–335 that there is no evidence of cracks in olivine grains. This is at best sloppy. Furthermore, why are these cracks forming, if not by the very reaction induced cracking that the authors claim is not observed?

Thanks for pointing this out. The sentence (lines 333-335 in the original version) was modified:

"Indeed, the nanotomography data show only dissolution features such as etch pits and worm holes, with no evidence of crystallization pressure induced cracks (Figure 11)" (lines 368-370)

Lines 278–288, and figure 12: It would be interesting to see plotted the individual volume change of the olivine grain and the formed precipitates. I would also suggest changing the figure a bit, it is hard to see the structure of the precipitated material. Something closer to figure 5 in Zhu et al. would be better.

We made diligent attempts to segment the different solid phases (i.e., olivine vs. precipitates). Unfortunately, the phase contrast between the precipitants and olivine grains is very small, and at the current spatial resolution of ~2 microns, we could not segment precipitants from olivine grains with acceptable uncertainties. Even at the sites where large orthorhombic crystals are present, it is difficult to determine the phase boundaries between olivine and orthorhombic crystals. Improved imaging techniques and perhaps different experimental designs are needed to quantify the reaction progress.

We modified the figure (now Figure 13) to increase the contrast between olivine and precipitates.

*Figure 1: The text "Reacted" is extremely hard to see, please consider changing the color.*

Thanks for pointing this out. We have changed the color in the revised Figure 15.

**Figure 5: Why is the dissolution or crack always in the lower right corner?**

This is a mere coincidence. Dissolution features and cracks are also observe in other part of the sample. Figures below shows that this dissolution is also observed in other places rather than just the lower right corner.